# A threonyl-tRNA synthetase-mediated translation initiation machinery

Seung Jae Jeong[1,2], Shinhye Park[3], Loi T. Nguyen[3,4], Jungwon Hwang [3], Eun-Young Lee[3], Hoi-Khoanh Giong[5,6,7], Jeong-Soo Lee[5,6], Ina Yoon[1,2], Ji-Hyun Lee[1], Jong Hyun Kim[1], Hoi Kyoung Kim[1], Doyeun Kim[1], Won Suk Yang[1], Seon-Young Kim [7,8], Chan Yong Lee[4], Kweon Yu [5,6,7], Nahum Sonenberg[9,10], Myung Hee Kim[3] & Sunghoon Kim [1,2]

A fundamental question in biology is how vertebrates evolved and differ from invertebrates, and little is known about differences in the regulation of translation in the two systems. Herein, we identify a threonyl-tRNA synthetase (TRS)-mediated translation initiation machinery that specifically interacts with eIF4E homologous protein, and forms machinery that is structurally analogous to the eIF4F-mediated translation initiation machinery via the recruitment of other translation initiation components. Biochemical and RNA immunoprecipitation analyses coupled to sequencing suggest that this machinery emerged as a gain-of-function event in the vertebrate lineage, and it positively regulates the translation of mRNAs required for vertebrate development. Collectively, our findings demonstrate that TRS evolved to regulate vertebrate translation initiation via its dual role as a scaffold for the assembly of initiation components and as a selector of target mRNAs. This work highlights the functional significance of aminoacyl-tRNA synthetases in the emergence and control of higher order organisms.

[1] Medicinal Bioconvergence Research Center, Seoul National University, Suwon 16229, Korea. [2] College of Pharmacy, Seoul National University, Seoul 08826, Korea. [3] Infection and Immunity Research Laboratory, Metabolic Regulation Research Center, Korea Research Institute of Bioscience and Biotechnology (KRIBB), Daejeon 34141, Korea. [4] Department of Biochemistry, Chungnam National University, Daejeon 34134, Korea. [5] Disease Target Structure Research Center, KRIBB, Daejeon 34141, Korea. [6] Dementia DTC R&D Convergence Program, Korea Institute of Science and Technology, Seoul 02792, Korea. [7] KRIBB School of Bioscience, Korea University of Science and Technology, Daejeon 34141, Korea. [8] Personalized Genomic Medicine Research Center, KRIBB, Daejeon 34141, Korea. [9] Department of Biochemistry, McGill University, Montreal, Quebec H3A 1A3, Canada. [10] Rosalind and Morris Goodman Cancer Research Centre, Montreal, Quebec H3A 1A3, Canada. These authors contribute equally: Seung Jae Jeong, Shinhye Park. Correspondence and requests for materials should be addressed to M.H.K. (email: mhk8n@kribb.re.kr) or to S.K. (email: sungkim@snu.ac.kr)

Canonical cap-dependent translation is maintained by the mammalian target of rapamycin complex 1 (mTORC1) that phosphorylates eukaryotic translation initiation factor (eIF) 4E1-binding proteins (4E-BPs) and inhibits the interaction between 4E-BP and eIF4E1 (hereafter referred to as eIF4E) under normal cellular conditions[1]. Translation initiation begins with the recognition of the 7-methylguanosine (m7GpppN, where N is any nucleotide) 5′-cap structure of mRNAs by eIF4F, a heterotrimeric complex that is composed of the cap-binding protein eIF4E, the scaffold protein eIF4G1 (hereafter referred to as eIF4G), and the RNA helicase eIF4A1 (hereafter referred to as eIF4A)[2].

Cell condition-specific translation occurs in all eukaryotic lineages[3–7]. In humans, a different combination of eIF4 isoforms mediate cap-dependent translation initiation through canonical eIF4F inactivation, in which mTORC1 activity is repressed by a multitude of stresses and 4E-BP binds to and sequesters eIF4E[8]. For example, oxygen tension-specific translation initiation during hypoxia is triggered when the cap-dependent translation machinery switches from eIF4E to eIF4E2 (also known as eIF4E homologous protein, 4EHP), which assembles together with oxygen-regulated hypoxia-inducible factor 2α (HIF-2α) and RNA-binding protein RBM4 into a hypoxia-stimulated hetero-trimeric complex that regulates global hypoxic protein synthesis[5]. These findings demonstrate the potential for fundamental complexity in protein synthesis via alternative translation machineries, but detailed molecular mechanisms remain poorly understood.

eIF4E2 (hereafter referred to as 4EHP) is generally considered unlikely to stimulate translation initiation. Analysis of *Drosophila* 4EHP revealed that translational repression of *caudal* mRNA is required for embryogenesis[4,9]. 4EHP binds directly to the caps of both *caudal* mRNA and Bicoid protein, which tethers the 3′ untranslated region (UTR) of *caudal* mRNA to repress mRNA translation[4]. In mammals, 4EHP forms a complex with Grb10-interacting GYF protein 2 (GIGYF2) and the zinc finger protein 598 to repress translation of mRNAs during embryonic development[10]. Together, these studies suggest that 4EHP may act independently of eIF4E as a nexus for specific translation to orchestrate key cellular processes, both positively and negatively, in a binding partner-dependent manner.

One of the most fundamental questions in biology is how vertebrates evolved and differ from invertebrates. Vertebrates engage in specific and committed translational processes over and above those in invertebrates, and this reflects their greater complexity. Although many studies have focused on differences in genetic constitution and transcription, relatively little is known about differences in the regulation of translation in the two systems. In this study, we identified aminoacyl-tRNA synthetase-mediated cap-dependent vertebrate-specific translation initiation machinery and investigated its structure, function, and molecular mechanism. The machinery is analogous to eIF4F composed of the scaffold protein threonyl-tRNA synthetase (TRS), 4EHP, and eIF4A. TRS exhibits dual functionality to determine the vertebrate specificity of the machinery via its unique N-terminal extension that represents a gain-of-function component, and its ability to select target mRNAs.

## Results

**Specific interaction of TRS with 4EHP.** In addition to its tRNA-charging activity, *Escherichia coli* TRS represses the translation of its own mRNA by binding to the 5′ UTR, which forms a pseudo-anticodon loop[11]. This observation inspired us to test the potential role of human TRS in the control of translation. To obtain mechanistic insight, we first identified cellular factors that associate with human cytosolic TRS by using affinity purification

mass spectrometry. Of the 434 proteins identified as potential direct or indirect TRS-interacting components, factors involved in post-transcriptional regulation (88), mRNA metabolic process (91), and translation (83) were enriched with high statistical significance (Supplementary Fig. 1a, b). We also subjected human TRS to yeast LexA-B42 two-hybrid screening using a HeLa cell cDNA library and identified four proteins as potential TRS interactors, including TRS itself (Supplementary Fig. 1c). Both analyses independently revealed that TRS interacts with 4EHP (eIF4E2); thus, we focused on the functional importance of this interaction in translation control.

Pull-down assays of TRS and eIF4E isoforms co-expressed in 293T cells showed that TRS interacts with 4EHP (eIF4E2), but not with eIF4E (eIF4E1) or eIF4E3 (Fig. 1a). Conversely, co-immunoprecipitation in 293T cells revealed a specific interaction between 4EHP and TRS, but not with other aminoacyl-tRNA synthetases (ARSs; Fig. 1b). Interaction between endogenous TRS and 4EHP was further confirmed by co-immunoprecipitation in embryonic lung WI-26 cells (Fig. 1c, d). Bimolecular fluorescence complementation (BiFC) analysis using Venus green fluorescent protein[12] yielded fluorescence only with co-transfection of TRS-VN (Venus N-domain) and 4EHP-VC (Venus C-domain) and not with other pairs (TRS-VN/eIF4E-VC, TRS-VN/eIF4E3-VC, or other ARS-VN/4EHP-VC), further demonstrating the specificity of the TRS-4EHP interaction (Fig. 1e).

To identify the TRS region responsible for interaction with 4EHP, we generated plasmids encoding different functional TRS domains based on the published structure (PDB ID 1WWT and PDB ID 1QF6; Fig. 1f). Co-immunoprecipitation analysis of interactions between each TRS domain and 4EHP revealed that only the N-terminal region (UNE-T, residues 1−80) co-precipitated with 4EHP (Fig. 1g). Isothermal titration calorimetry (ITC) analysis confirmed the direct interaction of TRS UNE-T with 4EHP, which was fitted to a 1:1 binding model with a $K_d$ of 2.48 μM (Fig. 1h). We further evaluated the interaction of full-length TRS, which forms a dimer via its catalytic domain[13], with 4EHP at the protein level using in vitro Strep-Tactin pull-down assays. Unlikely the isolated UNE-T, full-length TRS interacted weakly with 4EHP (Supplementary Fig. 1d), suggesting that the N-terminal UNE-T may be only partially exposed in the dimeric form of TRS in vitro. Formation of the cellular TRS-4EHP complex might involve a conformation change of the full-length TRS to fully expose UNE-T. Collectively, these results demonstrate the specific interaction of TRS with 4EHP via its UNE-T region.

**Structure of the TRS UNE-T and 4EHP complex.** To further elucidate the interaction between TRS and 4EHP at the molecular level and to gain an insight into the function of the TRS-4EHP complex, we determined the crystal structure of the UNE-T region (residues 30−74) complexed with 4EHP (residues 45−234) by optimizing crystallization to obtain crystals suitable for high-resolution X-ray diffraction data collection.

The final model includes residues Lys45 to Asp219 of 4EHP and Pro49 to Glu74 of TRS UNE-T (Fig. 2a and Supplementary Table 1). Since crystallization of the TRS UNE-T-4EHP complex was carried out in the absence of a 7-methyl GTP (m7GTP) cap analog bound to 4EHP, residues Pro69−Tyr78 and Ser220 −Val234 in 4EHP were disordered and hence not visible in the electron density map, as was the case in the structure of 4EHP complexed with 4E-BP1 or GIGYF1/2 without m7GTP[14,15]. Residues Gly30−Asn48 in TRS were also not observed in the electron density map; hence these flexible regions were not included in the final model. The TRS UNE-T region interacting with 4EHP adopts a canonical helix with additional short N- and

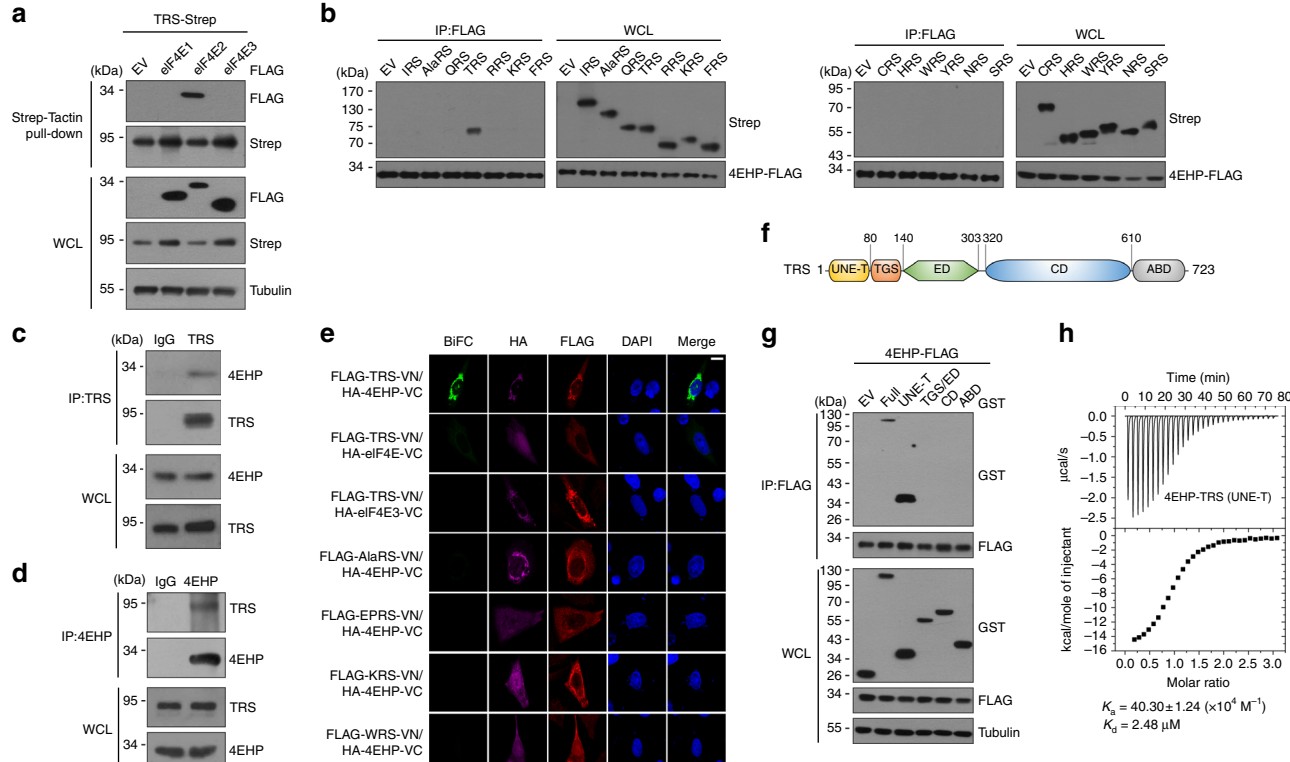

**Fig. 1** Human TRS interacts specifically with eIF4E2 (4EHP). **a** Pull-down assay of co-expressed TRS-Strep and eIF4E-FLAG isoforms in 293T cells. TRS-Strep was pulled down with Strep-Tactin beads, and co-precipitation of eIF4E isoforms was analyzed by immunoblotting with anti-FLAG antibody. **b** Immunoassay of co-expressed 4EHP-FLAG with human ARSs in 293 T cells. 4EHP-FLAG was immunoprecipitated with anti-FLAG antibody, and co-precipitation of ARSs-Strep was determined by immunoblotting with anti-Strep antibody. **c** Endogenous TRS immunoprecipitation with anti-TRS antibody in WI-26 cell lysates and co-immunoprecipitation of endogenous 4EHP determined using anti-4EHP antibody. Rabbit IgG was used as a negative control. **d** Co-immunoprecipitation of the above proteins confirmed in the reverse order. Rabbit IgG was used as a negative control. **e** Interactions between the indicated protein pairs determined by reconstitution of Venus green fluorescent protein (GFP) in Chinese hamster ovary (CHO) cells. Nuclei were stained with 4′,6-diamidino-2-phenylindole (DAPI; blue). Expression of HA-4EHP and FLAG-TRS was confirmed by immunofluorescence staining with anti-HA (Alexa 647; purple) and anti-FLAG (Alexa 594; red) antibodies, respectively. Scale bar = 10 μM. **f** Domain structure of human TRS determined based on the crystal and solution structures of *E. coli* (PDB 1QF6) and human (PDB 1WWT) TRS. **g** Immunoassay of co-expressed 4EHP-FLAG with full-length GST-TRS or the indicated domains in 293 T cells. Cell lysates were immunoprecipitated with anti-FLAG antibody, and co-precipitated TRS domain(s) were determined by immunoblotting with anti-GST antibody. **h** ITC determination of the binding affinity and stoichiometry of the TRS UNE-T region and 4EHP. Raw data and the integration plot are displayed in the upper and lower panel, respectively. Data are representative of at least three experiments, each with similar results (**a**−**e**, **g**, **h**). EV empty vector, WCL whole cell lysate, VN venus N-domain, VC venus C-domain, UNE-T unique region extension at the N-terminus of TRS, TGS a domain named after TRS, GTPase and SpoT, ED editing domain, CD catalytic domain, ABD anticodon-binding domain

C-terminal extensions (Fig. 2a), as previously reported for the complexes of 4E-BP1[16] and eIF4G[17] with eIF4E, and of 4E-BP1 and GIGYF1/2 with 4EHP[15]. Thus, the structure of the TRS UNE-T-4EHP complex superimposes well with those of 4E-BP1 or eIF4G complexed with eIF4E (Fig. 2b), and 4E-BP1 or GIGYF1/2 complexed with 4EHP (Fig. 2c).

Recent structural studies showed that eIF4G and 4E-BP1 interact with the dorsal and lateral surfaces of eIF4E via their canonical and non-canonical motifs, respectively[16,17] (Fig. 2b). Furthermore, GIGYF1/2 proteins and 4E-BP1 interact with 4EHP in the same manner that eIF4G and 4E-BP1 interact with eIF4E[15] (Fig. 2c). In addition to the common motifs, GIGYF1/2 proteins possess auxiliary sequences located immediately after these motifs that selectively bind to 4EHP and repress target mRNA expression[15] (Fig. 2c). However, TRS does not appear to utilize these non-canonical motifs and auxiliary sequences in the interaction with 4EHP because the TRS TGS (named after TRS, GTPase, and SpoT)[18], which does not interact with 4EHP (Fig. 1f, g), is stably structured immediately after Asp78 (PDB ID 1WWT). In addition, the non-canonical hydrophobic motif containing a strictly conserved phenylalanine[15–17] (Fig. 2b) and

well conserved auxiliary motif sequences[15] (Fig. 2c) located after the canonical motif in eIF4G, 4E-BP1 and GIGYF1/2, and GIGYF1/2, respectively are not found in the TGS domain.

Overall, the crystal structure of the TRS UNE-T-4EHP complex suggests that TRS may play a regulatory role in translation initiation via 4EHP, and this may be distinct from the previously reported eIF4E- or 4EHP-mediated regulation of translation initiation.

**Details of the interaction between TRS UNE-T and 4EHP.** The TRS UNE-T region engages the dorsal surface of 4EHP through its α-helix (Fig. 2c). The interaction is mediated via the canonical 4EHP-binding motif containing the canonical eIF4E-binding sequence YX₄Lφ (Y, X, L, and φ indicate Tyr, any amino acid, Leu, and any hydrophobic amino acid, respectively) with two N-terminal residues YX[19,20] (Fig. 2d, e). The canonical motif of UNE-T is located in a position similar to that in the complexes of 4E-BP1–4EHP[15], GIGYF1/2–4EHP[15], 4E-BP1-eIF4E[16], and eIF4G-eIF4E[17] (Fig. 2d, e).

Specifically, the hydroxyl group of TRS Tyr55 contacts the backbone carbonyl groups of His54 and Pro55 in the H54-P55-L56

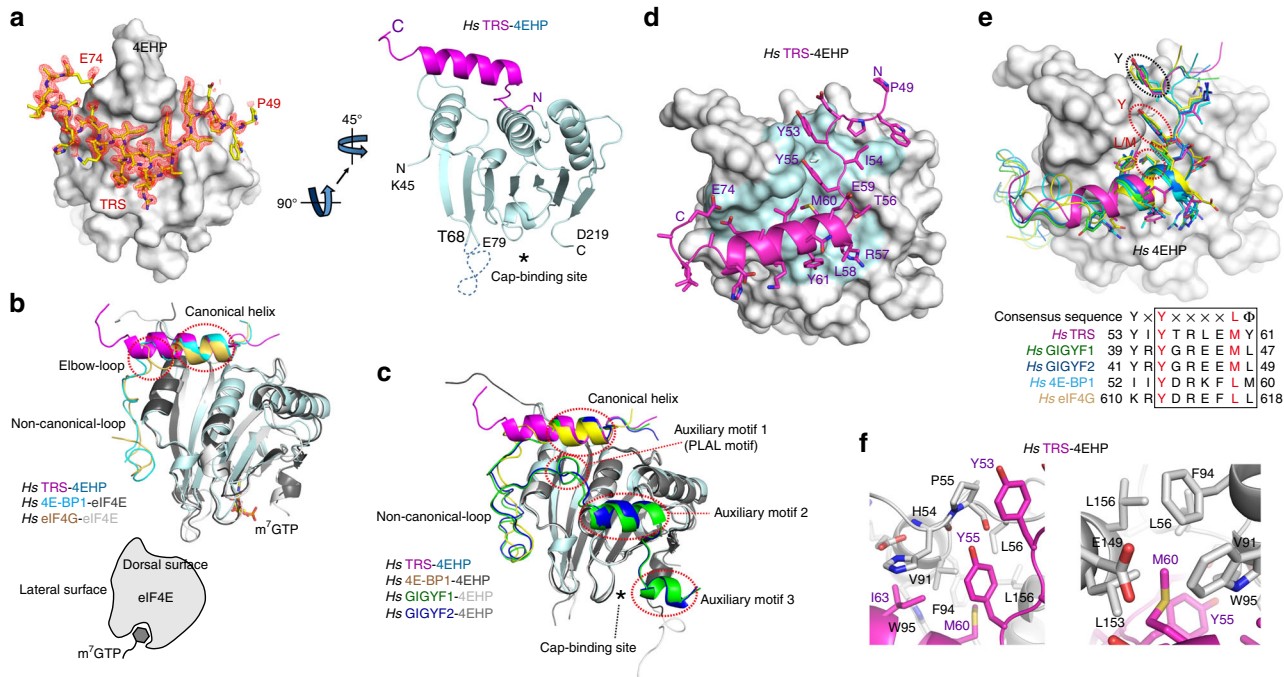

**Fig. 2** The structure of the TRS UNE-T region bound to 4EHP implies that it is part of the translation initiation complex. **a** Overall structure of the TRS UNE-T-4EHP complex. The $F_o - F_c$ electron density map of TRS UNE-T is displayed using surface representation of 4EHP calculated before the inclusion of TRS UNE-T, and contoured at 2.5 σ. The TRS UNE-T (magenta)-4EHP (pale cyan) complex is shown in cartoon representation. The disordered region in 4EHP (residues Pro69−Tyr78), which was not observed in the electron density map, is indicated by the dashed line. The asterisk indicates the cap-binding site in 4EHP. *Hs*, *Homo sapiens*. **b** Superimposition of the structures of the 4E-BP1 (cyan)-eIF4E (dark gray) and eIF4G (yellow)-eIF4E (light gray) complexes onto that of TRS (magenta) bound to 4EHP (pale cyan). The m$^7$GTP cap analog bound to eIF4E is shown in stick representation. Dorsal and lateral surfaces are indicated. **c** Superimposition of the structures of the 4E-BP1 (yellow)-4EHP (dark gray), GIGYF1 (green)-4EHP (light gray), and GIGYF2 (blue)-4EHP (gray) complexes onto that of TRS (magenta) bound to 4EHP (pale cyan). **d**, **e** Interactions between TRS and 4EHP mediated by the canonical eIF4E/4EHP-binding motif. TRS residues (magenta) involved in the interaction with 4EHP are shown in stick representation (**d**). Surface region (pale cyan) of 4EHP residues responsible for interactions with TRS (**d**). eIF4E/4EHP-interacting proteins superimposed on the TRS-4EHP complex (**e**). Sequence alignment of the eIF4E/4EHP-binding consensus motif. Y and L/M of the consensus sequences (YX$_4$Lϕ) and Y of the additional N-terminal residues (YX) are highlighted by red and black dotted circles, respectively (**e**). **f** Features of TRS Tyr55 (left) and Met60 (right) interacting with 4EHP

motif of 4EHP (corresponding to H37-P38-L39 in eIF4E) (Fig. 2f, left panel). The residue corresponding to Lϕ in the consensus sequence is substituted by Mϕ in TRS (Fig. 2d, e). In the structure, Met60 appears critical for interaction with 4EHP. It fits closely into the hydrophobic pocket formed by Val91, Phe94, Trp95, Leu153, and Leu156 of 4EHP (Fig. 2f, right panel) in a similar manner to that observed in GIGYF1/2 in complex with 4EHP[15].

Mutation of Tyr55 or Met60 in the TRS consensus sequence completely abolished co-immunoprecipitation with 4EHP, confirming their pivotal roles in the interaction with 4EHP (Supplementary Fig. 2a). A mild effect was observed for mutation of Tyr53 to Phe (Supplementary Fig. 2a) because the residue is surface-exposed and only marginally involved in the interaction with 4EHP (Fig. 2e, f). The same is true for its corresponding residues Tyr39 in GIGTYF1 and Tyr41 in GIGYF2[15]. Furthermore, mutation of 4EHP at His54, Phe94, and Leu156 abolished interaction with TRS (Supplementary Fig. 2b). The TRS M60K mutant retained enzymatic activity (Supplementary Fig. 2c) but lost its ability to bind to 4EHP (Supplementary Fig. 2a, d). Conversely, mutation of a catalytic residue in TRS (*e.g.*, C413S) ablated its enzymatic activity (Supplementary Fig. 2e) but did not affect binding to 4EHP (Supplementary Fig. 2d), suggesting that the two activities are mutually independent.

**Vertebrate-specific TRS-4EHP interaction.** While the catalytic domain of TRS is highly conserved throughout all three kingdoms, the UNE-T region is shared only among eukaryotic TRSs

(from yeast to human), suggesting that the TRS-4EHP interaction might be unique to eukaryotic organisms. Structure-based sequence alignment revealed that the 4EHP-interacting region containing the canonical α-helix in human TRS is similar to those of mouse and zebrafish TRSs but somewhat different from those of the yeast to fly enzymes (Fig. 3a), and this is also true for the 4EHP region containing the H-P-L motif involved in TRS binding (Fig. 3b). Co-immunoprecipitation confirmed the TRS-4EHP interaction in mouse (Fig. 3c) and zebrafish (Fig. 3d). Mutation of residues Asp50 and Ile55, corresponding to tyrosine and leucine, respectively, in the YX$_4$Lϕ motif of zebrafish TRS diminished the TRS-4EHP interaction (Fig. 3d). In contrast, fly TRS did not interact with either fly or human 4EHP (Fig. 3e). In addition, no interaction was observed between TRS and 4EHP from lower eukaryotic organisms (nematode and yeast; Fig. 3f, g). Interaction between TRS and 4EHP from the same species (human, zebrafish, and fly) and between fly TRS and human 4EHP was further confirmed by in vitro pull-down assay, and resulted in strong, modest, and no interaction for human, zebrafish, and fly pairs (Fig. 3h). Together, these results suggest a vertebrate-specific interaction between TRS and 4EHP.

**TRS selects mRNAs required for vertebrate development.** Our results suggest that the TRS-4EHP complex may play a role in translation initiation that is unique to the vertebrate lineage. To gain insight into the specificity of target mRNAs, we enriched mRNAs bound to TRS by immunoprecipitation with anti-TRS

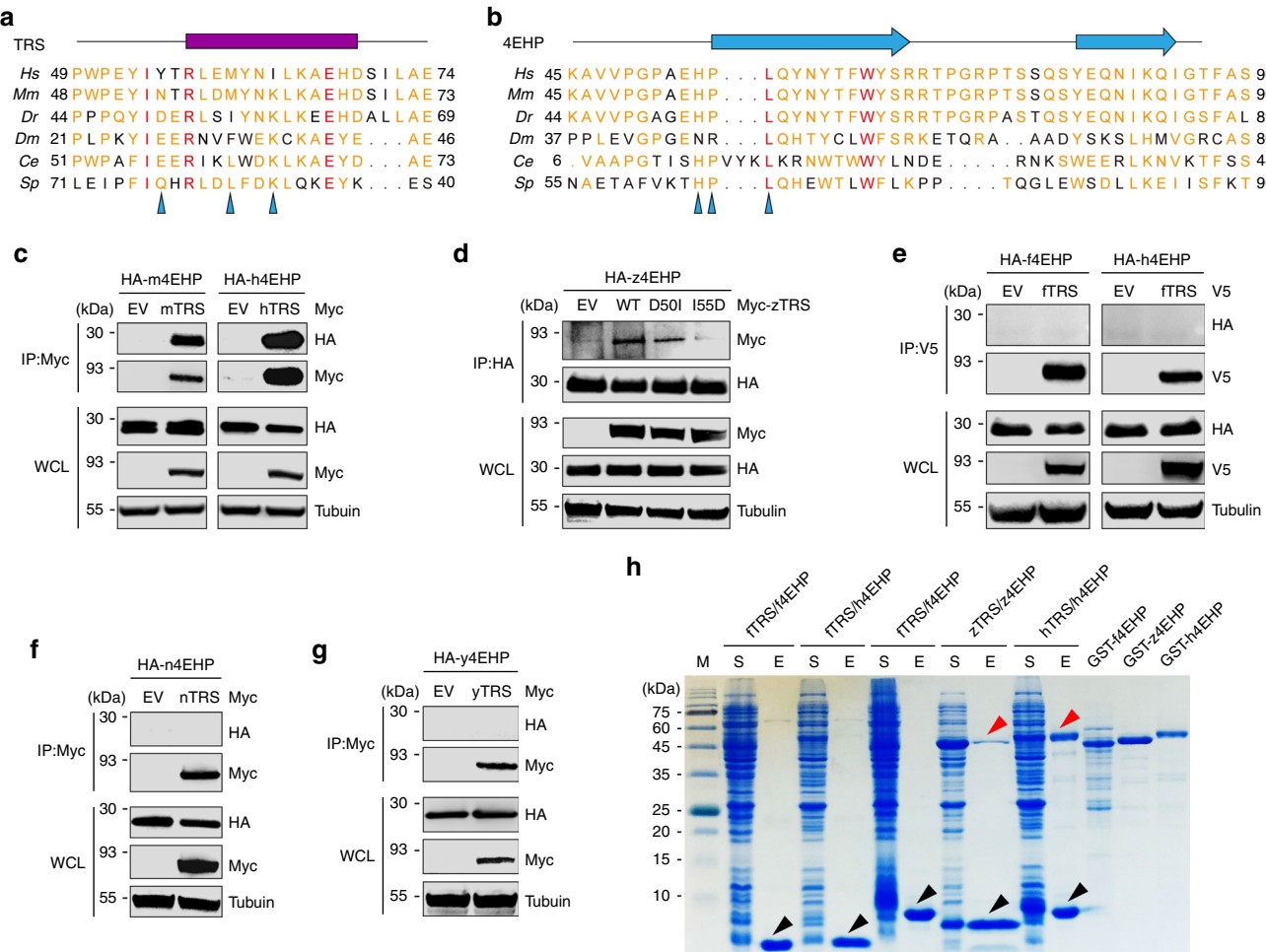

**Fig. 3** The TRS-4EHP interaction represents an evolutionary gain-of-function in vertebrates. **a**, **b** Structure-based sequence alignment of TRS and 4EHP in different species. Residues critical for the interaction between TRS and 4EHP are indicated by arrowheads. **c**−**g** Immunoassay of interactions between TRS and 4EHP pairs from the indicated species in 293T cell lysates (**c**, **d**, **f**, **g**) and *Drosophila* S2 cell lysates (**e**). Myc-TRS was immunoprecipitated with anti-Myc antibody, and co-precipitation of HA-4EHP was determined by immunoblotting with anti-HA antibody (**c**, **f**, **g**). HA-z4EHP was immunoprecipitated with anti-HA antibody, and co-precipitated zTRS WT or mutant (D50I, I55D) was determined by immunoblotting with anti-Myc antibody (**d**). V5-fTRS was immunoprecipitated with anti-V5 antibody, and co-immunoprecipitation of HA-f4EHP was detected by immunoblotting with anti-HA antibody (**e**). **h** In vitro pull-down assay of TRS (UNE-T)-His and GST-4EHP pairs from the indicated species. TRS-His was pulled down with Ni-NTA resin, and co-precipitated GST-4EHP was eluted from the resin and detected by Coomassie staining. Black and red arrowheads indicate eluted TRS (UNE-T)-His and GST-4EHP, respectively. Purified GST-fused 4EHP proteins from the indicated species are shown. Data are representative of at least three experiments, each with similar results (**c**−**h**). *Hs* Homo sapiens, *Mm* Mus musculus, *Dr* Danio rerio, *Dm* Drosophila melanogaster, *Ce* Caenorhabditis elegans, *Sp* Schizosaccharomyces pombe,. m mouse, h human, n nematode, y yeast, z zebrafish, f fly, EV empty vector, WCL whole cell lysate, M molecular size marker, S soluble fraction containing proteins expressed in *E. coli*, E proteins eluted from Ni-NTA resin

antibody, then performed RNA immunoprecipitation and sequencing (RIP-seq) and compared the mRNAs with those enriched by immunoprecipitation with IgG and anti-AlaRS antibody (Fig. 4a). We designated TRS as a determinant of target mRNAs in the RIP-seq experiments because 4EHP regulates translation initiation of diverse mRNAs. We identified 2,928 transcripts enriched in anti-TRS immunoprecipitates and subjected them to functional annotation clustering analysis. When classified by Gene Ontology (GO) terms in Biological Process categories, a large proportion (39.8%) of the 166 enriched GO terms were found to be related to system development (Fig. 4b). Remarkably, most genes in the system development group appear to reflect biological processes in vertebrates (*e.g.*, the development of nervous, skeletal, and circulation systems, and tube formation) that were important during the invertebrate-to-vertebrate transition (Fig. 4c). Additionally, we conducted mRNA enrichment experiments not only with anti-TRS antibody, but also with anti-

PRS and IRS antibodies, and compared the enriched mRNAs. Similar to the results obtained using AlaRS antibody as a control, functional annotation clustering analysis also yielded a large proportion of enriched GO terms related to vertebrate system development (Supplementary Fig. 3).

Among the TRS-enriched transcripts, we paid particular attention to genes involved in vasculogenesis and/or angiogenesis such as vascular endothelial growth factor (*VEGF*), which are considered evolutionary hallmarks of vertebrates. A previous study reported that the *E. coli* TRS anticodon-binding domain (ABD) selectively binds to an anticodon-like loop in the 5′-UTR of its own mRNA during auto-translational repression[11]. The *E. coli* TRS ABD is almost identical to its human counterpart[13], and the residues responsible for binding are strictly conserved between TRS ABDs in *E. coli* (e.g., Arg583, Glu600, and Arg609) and human (e.g., Arg663, Glu680, and Arg689). We therefore speculated that human TRS may interact with mRNAs

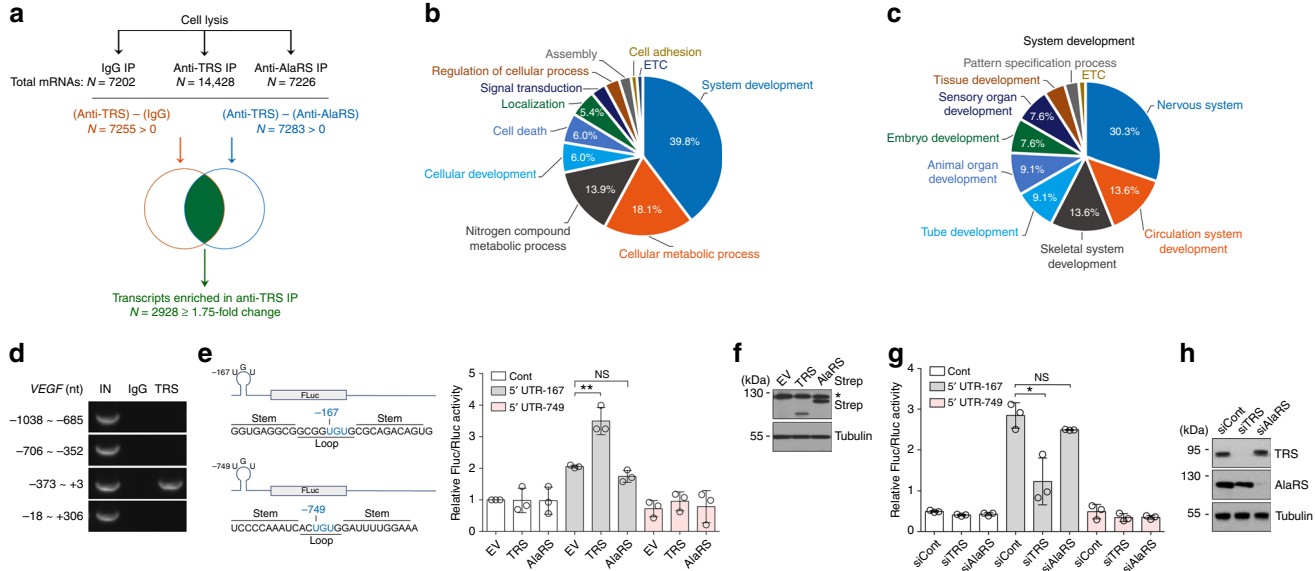

**Fig. 4** TRS-mediated translation initiation control of mRNAs in vertebrates. **a** Workflow used to identify TRS-targeted mRNAs. Total mRNAs isolated from 293 T cells were precipitated with anti-TRS or anti-AlaRS antibodies and/or mock IgG, and RNA sequencing of precipitated transcripts was conducted. TRS-enriched RNAs were subtracted from those enriched in AlaRS and IgG groups, and the two RNA pools were compared to identify common transcripts. Among the transcripts commonly detected in both TRS-enriched pools, 2,928 ≥1.75-fold were selected, and subsequently subjected to functional annotation. **b**, **c** Functional annotation of TRS-targeted mRNAs. Enriched GO terms in the Biological Process category were analyzed using the Database for Annotation, Visualization and Integrated Discovery (DAVID). **d** RNA immunoprecipitation of endogenous TRS in 293T cells followed by reverse transcription-PCR using primers within the 5′-UTR of *VEGF* mRNA. IN, input; nt, nucleotide. **e** Potential tRNA$^{Thr}$ anticodon triplet (UGU)-containing stem-loop structures were observed at positions -167 and -749 upstream from the *VEGF* mRNA initiation codon. The two RNA sequences spanning positions -1 to -540 (5′-UTR-167) and -541 to -1,038 (5′-UTR-749) were fused upstream of the *firefly* luciferase gene (*Fluc*) and co-expressed with the *Renilla* luciferase gene (*Rluc*) in TRS- or AlaRS-expressing 293T cells. Data are presented as the ratio of *firefly* to *Renilla* luciferase activity (Fluc/Rluc). **f** Immunoblot analysis of TRS-Strep and AlaRS-Strep in TRS- or AlaRS-expressing 293 T cells using Strep antibody. * indicates a nonspecific band. **g** Translation of the pseudo-anticodon-containing reporter gene in siTRS- or siAlaRS-transfected 293 T cells. siCont, non-targeting control siRNA. **h** The effect of knock-down of TRS or AlaRS with its specific siRNAs was determined by immunoblotting with each antibody. *$p < 0.05$; **$p < 0.01$; NS not significant vs. control group. Values are means ± SD of three independent experiments (**e**−**h**). EV empty vector

in a similar mode to *E. coli* TRS. To validate this possibility, we first assessed the region of *VEGF* mRNA in the 5′ UTR that interacts with TRS using RNA immunoprecipitation. The results revealed that TRS associates with the 5′ UTR of *VEGF* mRNA between nucleotides −373 and + 3 (Fig. 4d). We then searched for potential anticodon-like loop structures in the 5′-UTR of *VEGF* mRNA using RNA structure prediction[21], and a potential loop containing threonine anticodon-like bases (UGU) was identified at position −167 located in the TRS interaction region, and also at −749 (Fig. 4e). To evaluate whether these sites are responsive to the TRS-mediated translation initiation complex, we incorporated a 5′-UTR containing each of the anticodon-like base triplets (hereafter referred to as 5′-UTR-167 and 5′-UTR-749) upstream of the luciferase gene and expressed each in TRS- or AlaRS-expressing 293T cells. Luciferase expression was only increased with 5′ UTR-167 and not 5′ UTR-749 in TRS-expressing but not AlaRS-expressing 293 T cells (Fig. 4e, f). Changing the threonine anticodon UGU triplet to other bases decreased the efficiency of translation of the reporter gene (Supplementary Fig. 4a). Interestingly, changing UGU to another cognate anticodon sequence (CGU) did not affect the translation efficiency (Supplementary Fig. 4a), suggesting that TRS recognizes the anticodon$^{Thr}$-like loop structure of 5′ UTR in a similar manner to its binding to the anticodon of tRNA$^{Thr}$. In addition, both the loop position from the translation start site and the loop length appeared to be also critical factors for the recognition by TRS (Supplementary Fig. 4b, c). A potential TRS-binding loop was also identified at position −553 in the 5′-UTR of *ANG* mRNA, and similar results were obtained with the loop

incorporating the upstream region of the luciferase gene (Supplementary Fig. 4d, e).

Furthermore, translation of the reporter gene containing 5′ UTR-167 was markedly decreased in siTRS-transfected cells, but not in siAlaRS-transfected 293 T cells (Fig. 4g, h). Next, we tested the importance of residues Arg663, Glu680, and Arg689 that were predicted to bind the anticodon$^{Thr}$-like loop of the 5′ UTR during translation of the reporter gene. We substituted all of these residues with leucine, and translation was increased in TRS wild-type (WT)-expressing cells in a dose-dependent manner, but not in mutant-expressing cells (Supplementary Fig. 4f, g). These results indicate specificity of the TRS for selection of target mRNAs via interaction between the TRS ABD and the anticodon-like loop structure in the mRNA 5′ UTR.

**TRS and 4EHP-mediated vascular development**. We subsequently verified the functional significance of the TRS and 4EHP complex in vivo using cell and animal models. Suppression of TRS or 4EHP using specific siRNAs substantially downregulated VEGF at the protein level but not at the mRNA level, whereas eIF4E silencing had little effect (Supplementary Fig. 5a, b). A slight reduction in VEGF expression was observed when eIF4G was silenced (Supplementary Fig. 5a). Similar results were also obtained with angiogenin (ANG), the mRNA of which was also enriched with TRS (Supplementary Fig. 5c, d). Levels of secreted VEGF and ANG were subsequently diminished in various cell lines when TRS and/or 4EHP was suppressed (Fig. 5a, b). These results indicate that interaction between TRS and 4EHP positively regulates selective protein synthesis independently of eIF4E.

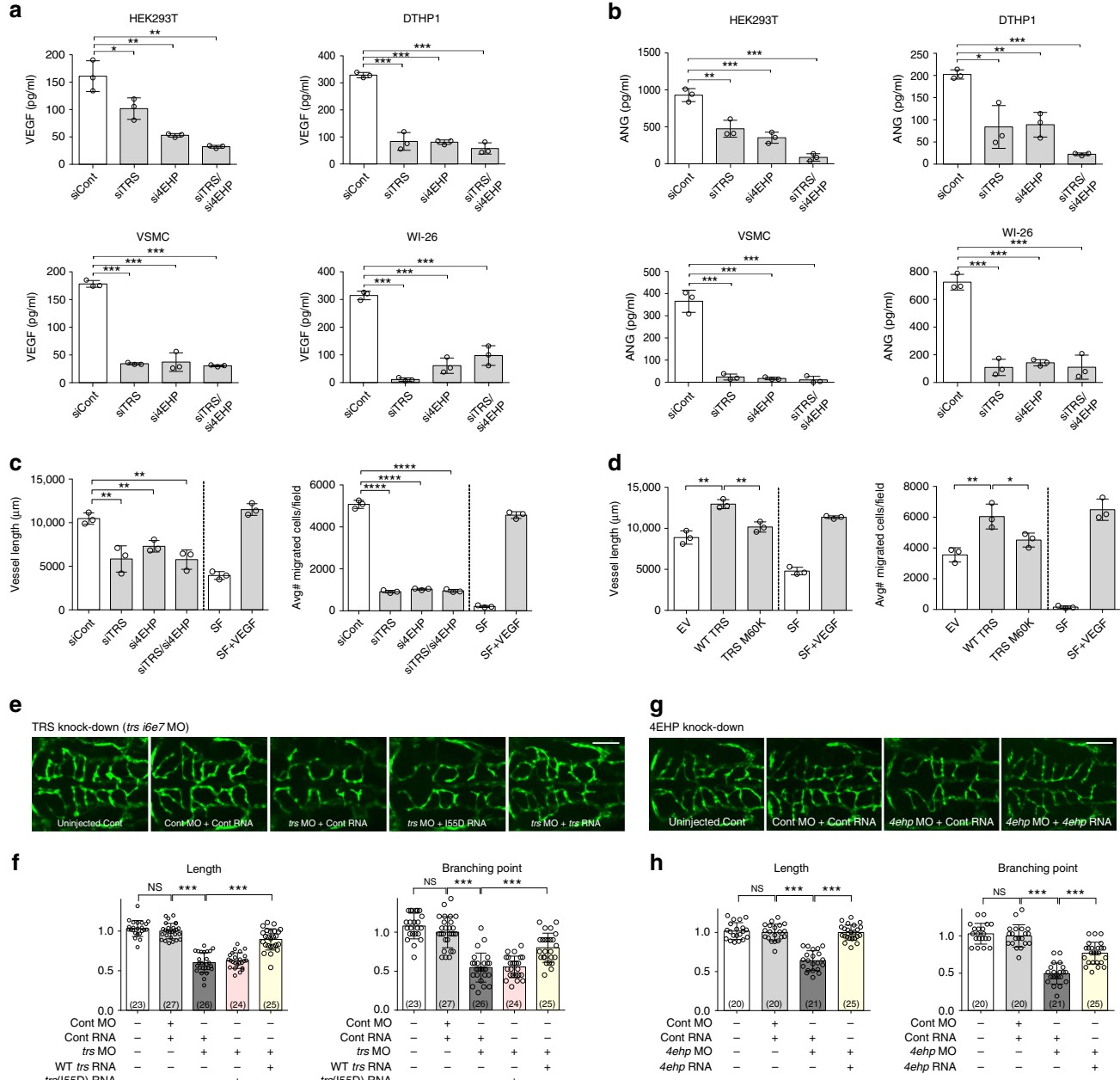

**Fig. 5** The 4EHP and TRS interaction is critical for translation initiation of mRNAs required for vascular development. **a** Effects of TRS and/or 4EHP on VEGF secretion in siTRS- and/or si4EHP-transfected cell lines. VEGF protein levels in the culture supernatant were determined by enzyme-linked immunosorbent assay (ELISA). **b** Effects of TRS and/or 4EHP on ANG secretion in siTRS- and/or si4EHP-transfected cell lines. ANG protein levels in the culture supernatant were determined by ELISA. **c**, **d** Effects of TRS and/or 4EHP on endothelial cell tube formation. Culture medium from WI-26 cells transiently transfected with siRNAs against TRS and/or 4EHP (or siCont) was used to treat HUVECs, which were subsequently plated on growth factor-reduced Matrigel to form capillary tubes (**c**). Total tube lengths in Supplementary Fig. 6a were measured using ImageJ (**c**). Migrated HUVECs were counted from three randomly selected fields (**c**). Culture medium from WI-26 cells transiently transfected with Myc-TRS WT or its 4EHP-binding-defective M60K mutant was used to treat HUVECs, which were subsequently plated on growth factor-reduced Matrigel to form capillary tubes (**d**). Total tube lengths in Supplementary Fig. 6f were measured using ImageJ (**d**). Migrated HUVECs were counted from three randomly selected fields (**d**). VEGF (10 ng mL$^{-1}$) served as a positive control. siCont, non-targeting control siRNA. *$p < 0.05$; **$p < 0.01$; ***$p < 0.001$; ****$p < 0.0001$ vs. control group. Values are means ± SD of three independent experiments (**a**–**d**). **e**–**h** TRS- and 4EHP-mediated translation initiation is critical for vascular development. TRS was suppressed using a morpholino (*trs i6e7* MO) together with control (Cont) RNA, or reconstituted with WT or I55D mutant *trs* RNA at 52 h post-fertilization (**e**). 4EHP was suppressed using a morpholino (*4ehp* MO) together with Cont RNA, or reconstituted with WT *4ehp* RNA (**g**). Quantitation and statistical analysis of the lengths and branching points of central arteries in the hindbrain of *Tg* (*kdrl:EGFP*) zebrafish embryos (**f**, **h**). Scale bar = 100 μM. The number of analyzed zebrafish embryos is shown in parentheses. ***$p < 0.001$; NS not significant vs. control group. Values are means ± SD. SF serum-free medium

Silencing of TRS or 4EHP had a comparable effect on VEGF protein and mRNA levels in both normoxia and hypoxia conditions (note that VEGF is intrinsically induced during hypoxia; Supplementary Fig. 5e, f), suggesting that TRS- and 4EHP-mediated translation initiation is not regulated by oxygen availability. It is worth mentioning that expression of TRS and 4EHP was not affected by oxygen tension (Supplementary Fig. 5e). VEGF translation is sensitive to the levels of both eIF4E and eIF4G, under both normal and hypoxic conditions[22]. Similarly, our results showed that silencing of eIF4E or eIF4G reduces VEGF translation under both normoxic and hypoxic conditions (Supplementary Fig. 5g).

We further evaluated whether TRS- and 4EHP-regulated translation influences endothelial cell migration and vessel (tube) formation using human umbilical vein endothelial cells (HUVECs). Culture supernatants from embryonic lung WI-26 cells treated with siTRS and/or si4EHP RNAs resulted in lower tube formation and endothelial cell migration than those from siCont- or VEGF-treated cells (Fig. 5c and Supplementary Fig. 6a, b). Similar results were also observed with supernatants from 293 T cells (Supplementary Fig. 6c−e). Tube formation and cell migration of HUVECs were significantly enhanced by supernatants from TRS-transfected WI-26 cells, but to a lesser extent by supernatants from 4EHP-binding-defective TRS (M60K)-expressing cells (Fig. 5d and Supplementary Fig. 6f, g). Since the secretion of catalytically active TRS induces angiogenesis[23], we measured the secretion of TRS and its M60K mutant protein, and the results revealed comparable secretion levels (Supplementary Fig. 6h), indicating that the difference in angiogenesis activity was not attributable to differences in secretion.

We next explored TRS- and 4EHP-mediated translation in a zebrafish model to investigate the closed circulatory system that is unique to vertebrates. The involvement of TRS or 4EHP in angiogenesis in central arteries (CtAs) in the hindbrain of developing zebrafish embryos (Supplementary Fig. 7a) was assessed by suppressing TRS or 4EHP using splice-blocking morpholinos (MOs; Supplementary Fig. 7b, c). Suppression of TRS using *trs i6e7* MO (*trs* MO + Cont RNA) decreased CtA length and branching points by 40% and 50%, respectively, compared with controls (Cont MO + Cont RNA; Fig. 5e, f). Defective angiogenesis caused by TRS suppression was rescued by expressing TRS (*trs* MO + *trs* RNA) but not the 4EHP interaction-defective I55D mutant (*trs* MO + I55D RNA; Fig. 5e, f). Similarly, 4EHP suppression (*4ehp* MO + Cont RNA) reduced CtA length and branching points by 36% and 50%, respectively, and these defects were also rescued by expression of 4EHP (*4ehp* MO + *4ehp* RNA; Fig. 5g, h). In addition, several angiogenic defects including vessel shortening and mis-sprouting of intersegmental vessels (ISVs) in the trunk of zebrafish embryos at the same stage were also observed upon 4EHP suppression (Supplementary Fig. 7d). Consistent with their roles in angiogenesis, TRS and 4EHP were strongly expressed in the developing trunk and hindbrain (Supplementary Fig. 8). Taken together, these results demonstrate that TRS and 4EHP-mediated translation is crucial for vascular development, and further support the notion that it is selective for biological processes related to system development unique to vertebrates.

**Functional similarity of TRS to eIF4G.** 4EHP is an mRNA 5′-cap structure-binding protein[14,24] that regulates translation of a subset of mRNAs[4,5,10,25], and 4EHP-interacting partners are believed to dictate its molecular and physiological functions[10]. Accordingly, we questioned how the TRS-4EHP complex positively controls translation initiation mechanically. To address this question, we first checked the dependency of the cap on the

interaction between the two proteins. Both proteins were detected in cap analog m7GTP-Sepharose precipitates, but not in control Sepharose precipitates, in all cell lines tested (Supplementary Fig. 9), indicating that TRS associates with cap-bound 4EHP.

The most critical event in typical cap-dependent translation is the formation of the eIF4F complex consisting of eIF4E, eIF4G, and eIF4A. Since the TRS-4EHP complex resembles the eIF4G-eIF4E complex and positively regulates translation, we examined the potential interaction of TRS with eIF4A by pull-down assays. TRS evidently interacted with eIF4A, but not with eIF4G (Fig. 6a), and interaction between TRS and eIF4A was maintained in the absence of 4EHP, and even when the interaction of TRS with 4EHP was disrupted by mutation (Fig. 6b). We further assessed the interaction of TRS and eIF4A in vitro. Purified TRS-His and eIF4A were incubated with eIF4A antibody and pulled down with Protein A/G PLUS-agarose beads. Co-purification of the two proteins as a complex was confirmed by immunoblotting analysis, showing that TRS directly interacts with eIF4A (Supplementary Fig. 10a). Co-immunoprecipitation assays further showed that TGS and the editing domain (TGS/ED) and the ABD of TRS are responsible for binding to eIF4A (Fig. 6c). However, our results showed that the TRS ABD site critical for binding to the anticodon[Thr]-like loop does not overlap with the region interacting with eIF4A (Supplementary Fig. 10b).

Together, the above results suggest that TRS may assemble a vertebrate-specific eIF4F-like complex together with 4EHP and eIF4A, similarly to eIF4G in the eIF4F complex. To validate this hypothesis, we performed pull-down assays of endogenous translation initiation factors, and observed precipitation of eIF4G, eIF4E, eIF4A, TRS, and 4EHP with m7GTP-Sepharose in 293 T cells (Fig. 6d). To investigate the relationship between the TRS-4EHP and eIF4G-eIF4E complexes, we monitored how suppression of each complex affects the formation of the other complex. When TRS was silenced, eIF4G, eIF4E, and eIF4A (the components of the eIF4F complex) were pulled down with m7GTP-Sepharose (Fig. 6d), and a small amount of 4EHP was detected, perhaps due to its intrinsic affinity for the m7GTP cap. However, when 4EHP was suppressed, only the eIF4F components were detected at significant levels (Fig. 6d). Next, we examined the effects of eIF4G and eIF4E suppression on the formation of the TRS-mediated complex. When eIF4G or eIF4E was suppressed, TRS and 4EHP were mainly detected with the cap (Fig. 6e). It should be noted that eIF4E binds to the cap analog m7GTP with approximately 100-fold greater affinity than 4EHP[24]. Thus, eIF4E was still detected even when eIF4G or eIF4E was silenced. Since eIF4A is commonly associated with both TRS-4EHP and eIF4E-4G complexes, suppression of one complex did not appear to completely prevent eIF4A binding to the other intact complex. Thus, its association with either complex was somewhat diminished compared with the other components (Fig. 6d, e). These results suggest that TRS acts as an eIF4G-like scaffold and represents an eIF4F analog that acts independently of eIF4F.

We separately suppressed TRS, 4EHP, eIF4G, and eIF4E using specific siRNAs, and the resulting cell lysates were immunoblotted with anti-puromycin antibody to monitor the effect of suppression on de novo protein synthesis as previously described[26]. Suppression of TRS and 4EHP did not influence global translation under these experimental conditions (Supplementary Fig. 11a, b). The absence of a significant effect on global translation caused by TRS knock-down was further validated by 35S Met-incorporation assays (Supplementary Fig. 11c). Silencing of eIF4E reduced cellular protein synthesis, and translation was more sensitive to eIF4G (Supplementary Fig. 11a, b) as previously reported[5,27].

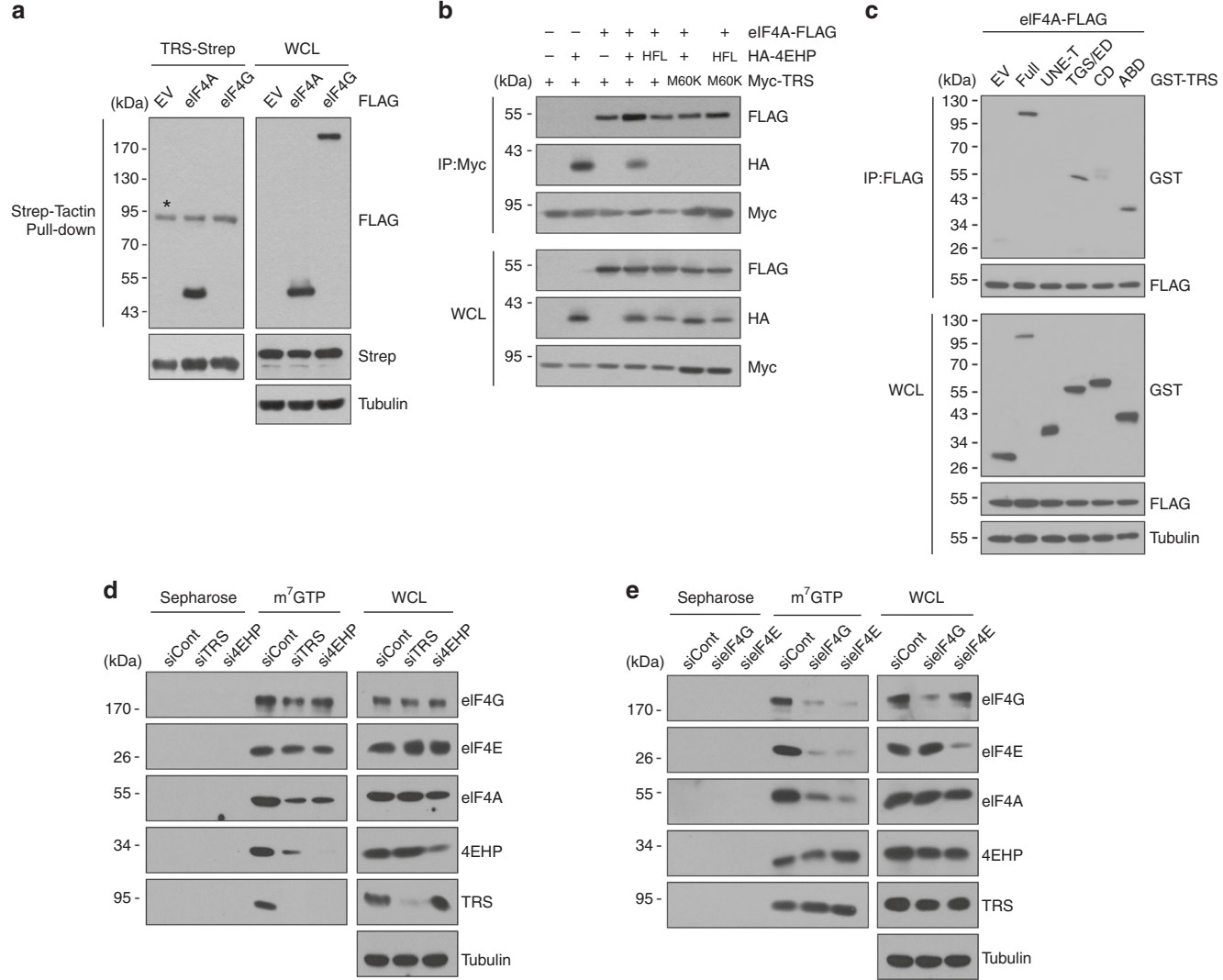

**Fig. 6** TRS functions similarly to eIF4G and acts as an eIF4F analog. **a** Pull-down assay of co-expressed TRS-Strep with eIF4A- or eIF4G-FLAG in 293 T cells. TRS-Strep was pulled down with Strep-Tactin beads, and co-precipitation of eIF4A or 4 G was determined by immunoblotting with anti-FLAG antibody. EV, empty vector. * indicates a nonspecific band. **b** Immunoassay of the co-expression of different combinations of plasmids in 293T cells. Myc-TRS was immunoprecipitated with anti-Myc antibody, and co-precipitation of other proteins was determined using tag-specific antibodies. **c** Immunoassay of co-expressed eIF4A-FLAG with GST-fused full-length TRS or its various domains in 293T cells. eIF4A-FLAG was immunoprecipitated with anti-FLAG antibody, and co-precipitated TRS proteins were determined by immunoblotting with anti-GST antibody. **d** Pull-down assay of endogenous translation initiation factors with m7GTP-Sepharose beads in 293 T cells transfected with siRNAs against TRS, 4EHP, or a non-targeting control (siCont). Cap-bound proteins were eluted from beads and immunoblotted with the indicated antibodies. Sepharose beads were used as a negative control. **e** Pull-down assay of endogenous translation initiation factors with m7GTP-Sepharose beads in 293T cells transfected with siRNAs against eIF4G, eIF4E, or siCont, and their suppression effects on cap-binding of other components, were determined as in (**d**). The data are representative of at least three experiments, each with similar results

**TRS-mediated translation initiation machinery**. The poly(A)-binding protein (PABP) interacts directly with eIF4G to facilitate translation initiation of polyadenylated mRNAs[28,29]. We therefore tested whether PABP is also associated with the TRS-mediated translation initiation complex. Pull-down assays showed that TRS indeed interacts with PABP in addition to 4EHP and eIF4A (Fig. 7a). RNase treatment did not alter the interaction between the two proteins, further supporting the direct interaction of TRS and PABP (Supplementary Fig. 12a). Co-immunoprecipitation assays further demonstrated that PABP interacts with the 4EHP-binding defective M60K mutant, indicating that the interaction is independent of 4EHP (Fig. 7b), and that the TRS ABD is responsible for binding to PABP (Fig. 7c). However, the TRS region interacting with PABP does not overlap

with the ABD site that is critical for binding to the anticodon$^{Thr}$-like loop (Supplementary Fig. 10b). To verify the direct interaction, purified TRS-Strep and His-GST-PABP were co-eluted from Strep-Tactin resin and subjected to immunoblotting analysis. The results revealed that the two proteins directly interact with each other (Supplementary Fig. 12b), and interactions between endogenous TRS, eIF4A, and PABP were further confirmed by co-immunoprecipitation (Fig. 7d).

In mammals, the direct interaction between eIF4G and eIF3 acts as a bridge between the mRNA cap-binding complex and the 40S ribosomal subunit to initiate substantial translation[30]. Given this knowledge, we examined whether TRS interacts with eIF3 and, if so, how it may be linked to eIF3. We first conducted in vitro pull-down assays of GST-TRS with each of the

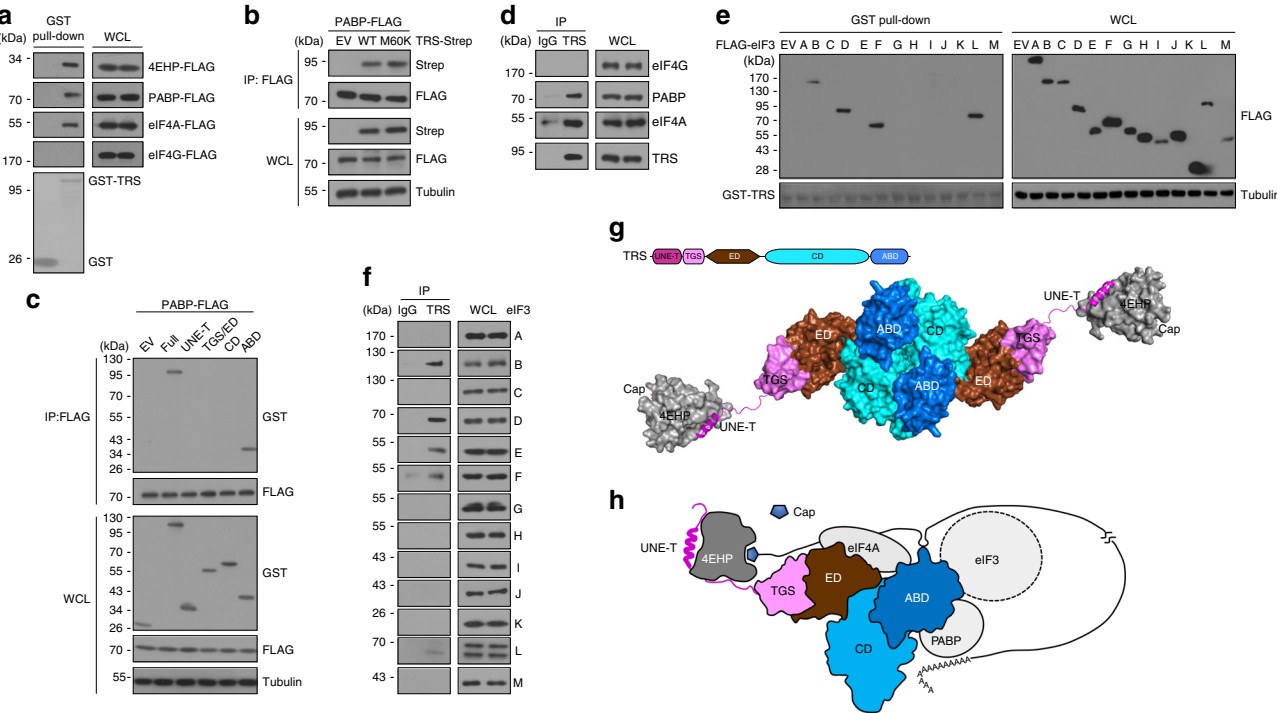

**Fig. 7** Discovery of a TRS-mediated translation initiation machinery. **a** Pull-down assay of GST-TRS with FLAG-tagged proteins. Purified GST-TRS from *E. coli* was incubated with 293T cell lysates with FLAG-tagged 4EHP, PABP, eIF4A, or eIF4G. GST-TRS was pulled down with glutathione-Sepharose beads, and co-precipitation of each FLAG-tagged protein was determined by immunoblotting with anti-FLAG antibody. **b** Immunoassay of co-expressed PABP-FLAG with WT or M60K mutant TRS-Strep in 293T cells. PABP-FLAG was immunoprecipitated with anti-FLAG antibody, and co-precipitated TRS proteins were determined by immunoblotting with anti-Strep antibody. **c** Immunoassay of co-expressed PABP-FLAG with GST-fused full-length TRS or its various domains in 293T cells. PABP-FLAG was immunoprecipitated with anti-FLAG antibody, and co-precipitated TRS proteins were determined by immunoblotting with anti-GST antibody. **d** Endogenous TRS was immunoprecipitated from WI-26 cells with a mouse anti-TRS antibody, and co-immunoprecipitates were examined with the indicated antibodies. Mouse IgG was used as a negative control. **e** Pull-down assay of GST-TRS with FLAG-tagged eIF3 subunits. Purified GST-TRS from *E. coli* was incubated with 293T cell lysates containing each of the FLAG-tagged eIF3 subunits. GST-TRS was pulled down with glutathione-Sepharose beads, and co-precipitation of each FLAG-tagged protein was determined by immunoblotting. **f** Endogenous TRS was immunoprecipitated from WI-26 cells with a mouse anti-TRS antibody, and co-immunoprecipitates were examined with the indicated antibodies. Mouse IgG was used as a negative control. Data are representative of at least three experiments, each with similar results (**a**−**f**). **g** Structural model showing the interaction of 4EHP with the UNE-T region of the TRS dimer. Each domain of TRS is labeled as in Fig. 1f. A monomer of TRS is shown in surface representation, and the UNE-T regions interacting with both 4EHP molecules are shown in cartoon representation. The model was built based on the crystal structure (PDB 1QF6) of *E. coli* TRS[56]. **h** Schematic model of the TRS-mediated translation initiation machinery. See the main text for a full description. Although TRS is predicted to function as a homodimer, the model only shows the monomer for simplicity. eIF3 is depicted by a dotted circle because its binding region in TRS has not yet been determined

eIF3 subunits using glutathione-Sepharose, and found that eIF3 subunits B, D, F, and L co-precipitated with GST-TRS (Fig. 7e). We then investigated the interaction between endogenous TRS and eIF3, and consistent with the in vitro pull-down assay, TRS interacted with the same subunits, as well as subunit E (Fig. 7f). Like eIF4A and PABP, the eIF3 subunit F-interacting site in TRS does not overlap with the region that binds to the anticodon[Thr]-like loop (Supplementary Fig. 10b). Of note, our interactome data also revealed association between eIF3 subunits and TRS (Supplementary Fig. 1b). These results indicate that TRS links the TRS-4EHP complex with eIF3 during translation initiation.

Based on these results, we propose a schematic model for the TRS-mediated translation initiation machinery. As a class II-type tRNA synthetase[31], human TRS forms a dimer via its catalytic domain[13], and our ITC results showed that TRS UNE-T binds to 4EHP with 1:1 stoichiometry (Fig. 1h). Accordingly, 4EHP may interact with each TRS monomer via the UNE-T region (Fig. 7g). It is noteworthy that the linker region between the UNE-T helix and the TGS domain of TRS is likely to be flexible (Fig. 7g). In the model of a single TRS unit (Fig. 7h), TRS specifically interacts

with 4EHP via its canonical helix motif, followed by recruitment of eIF4A to form the eIF4F-like complex and induces 4EHP binding to the cap located at the 5′-UTR of target mRNAs. The TRS ABD determines the selectivity of mRNAs by mimicking tRNA anticodon binding. PABP then integrates into the complex via interaction with the TRS ABD to form the efficient translation initiation machinery[32]. The molecular details underlying how the TRS ABD associates with several translation initiation components requires further investigation. In addition, whether recognition of the anticodon-like loop by the TRS ABD is the driving force for recruiting translation initiation factors should be further assessed in future work. Eventually, TRS links the complex to the ribosome via interaction with eIF3 to promote translation initiation. Given that native TRS exists as a dimer, it would be interesting to see whether the dimer interface provides an additional structural feature to accommodate other translation factors.

## Discussion

Expansion in molecular signaling complexity in accordance with the divergence of vertebrates from invertebrates required

significant changes in translation and its regulation. Herein, we identified a translation initiation machinery resulting from an evolutionary gain-of-function that is unique to vertebrates. ARSs are thought to have first appeared during the emergence of life as part of the essential process of decoding genetic information into proteins. Higher eukaryotic ARSs have undergone stepwise molecular evolution, including the addition of new domains with unique structural features that correlate with the increasing biological complexity of higher organisms. Among these new additions, UNE regions sharing no detectable sequence similarity with other structural modules of ARSs[33] are believed to endow unique functions to the attached enzymes[34]. Several UNEs including UNE-S at the C-terminus of SRS and UNE-I$_2$ at the C-terminus of isoleucyl-tRNA synthetase are present only in vertebrates[34]. UNE-S contains a nuclear localization signal and bestows SRS with a novel nuclear function in vascular development of vertebrates[35].

Interestingly, mammalian ARSs also appear to control translation via multiple routes. For instance, DNA damage-activated general control nonrepressed-2 kinase phosphorylates methionyl-tRNA synthetase at the critical tRNA-binding site to inactivate its methionine-charging activity, which slows down nascent protein synthesis until the damaged DNA is repaired[36]. Another well-known example is glutamyl-prolyl tRNA synthetase that is involved in the heterotetrameric interferon-γ-activated inhibitor of translation (GAIT) complex. The enzyme binds to the structural GAIT elements in the 3′ UTRs of multiple inflammation-related mRNAs and represses their translation.

In the present study, we discovered that TRS regulates vertebrate-specific translation initiation. In addition to its unique region, it engages in evolutionarily conserved characteristic binding to eukaryotic initiation factors involved in ribosome recruitment, as well as mRNA, and these multiple activities render it optimally adapted for specific translational regulation. Together, these results suggest that ARSs may play critical roles in the development of physiological systems and the maintenance of translational homeostasis in cells and organisms beyond their intrinsic functions as enzymes.

Vertebrate cardiovascular systems are considerably more complex than those of invertebrates, and vessel formation in the developing lung requires embryonic cardiovascular development[37]. Early endothelial cell differentiation followed by migration and tube formation is observed at embryonic day 10.5 during murine embryonic development, along with pulmonary vessel formation[38]. Coincidently, we also observed the emergence of murine 4EHP and VEGF at embryonic day 10.5 (Supplementary Fig. 13), implying that expression of 4EHP might be a cue for TRS to assemble a specific translation initiation complex. The precise mechanism underlying the temporal and spatial collinearity of TRS and 4EHP in translation initiation is worthy of further exploration. Our findings throw light on the evolutionary gain-of-function in translation that contributed to the divergence of vertebrates from invertebrates. Whether other specific translation processes are involved in the increased complexity in the vertebrate system relative to its invertebrate counterpart requires further investigation.

## Methods

**Cell culture**. HEK293T and WI-26 cells were grown in Dulbecco's Modified Eagle's Medium (DMEM; Hyclone) containing 10% fetal bovine serum (FBS; Hyclone) and antibiotics. Cell lines were cultured in 5% $CO_2$ at 37 °C. Chinese hamster ovary (CHO) and THP1 cells were grown in RPMI 1640 (Hyclone) containing 10% FBS and antibiotics. THP1 cells were differentiated to the adherent macrophage-like state by treatment with 50 ng mL$^{-1}$ phorbol 12-myristate 13-acetate (Sigma) for 36 h. Adhered differentiated cells were washed with cold phosphate-buffered saline (PBS) every 24 h for the next 3 days, and used as differentiated THP1 (DTHP1) cells. *Drosophila* Schneider (S2) cells were propagated at 28 °C in Schneider's

*Drosophila* medium (Life Technologies) supplemented with 10% FBS, 50 units mL$^{-1}$ penicillin, and 50 μg mL$^{-1}$ streptomycin. Human brain vascular smooth muscle cells (VSMCs) were maintained in SMCM (ScienCell) and used between 2 and 4 passages. HUVECs were maintained in EGM-2 complete media (Lonza) and used between 5 and 9 passages. Hypoxia was induced by incubating cells at 37 °C in a 1% $O_2$, 5% $CO_2$, and $N_2$-balanced atmosphere for 24 h.

**DNA cloning**. Human eIF4E1, eIF4E2, eIF4A1, eIF4G1, PABP, eIF3C, eIF3D, eIF3E, eIF3F, eIF3G, eIF3H, eIF3I, eIF3J, eIF3K, eIF3L, and eIF3M were cloned into pIRES-FLAG[39]. Human eIF4E3, and eIF3A, and eIF3B genes were purchased from Origene. The functional domains of human TRS were cloned into pEBG-GST (Addgene). 4EHP (eIF4E2) and TRS from human, mouse, zebrafish, fly, and nematode were cloned into pcDNA3.1-HA (Invitrogen) and pcDNA3.1/Myc-His A (Invitrogen), respectively. 4EHP and TRS from yeast were cloned into pCMV-HA (Clontech) and pcDNA3.1/Myc-His A, respectively. 4EHP from human, and 4EHP and TRS from fly were cloned into pAc5.1/V5-His A (Invitrogen). Human TRS was cloned into pEXPR-IBA103 (IBA) and pEXPR-IBA105 (IBA), and human AlaRS, CRS, FRS, HRS, IRS, KRS, NRS, QRS, RRS, SRS, WRS, and YRS were cloned into pEXPR-IBA103. All luciferase constructs were generated by cloning the respective PCR fragment into the pGL2 luciferase reporter (Promega). To produce the stable human 4EHP in complex with human TRS, a plasmid co-expressing the two proteins was constructed using a dual promoter vector system[40]. In this construct, the N-terminal extension region of TRS (residues 30−74) and 4EHP (residues 45−234) were independently expressed under the control of the T7 promoter and the Tac promoter, respectively. Briefly, the gene encoding 4EHP (residues 45−234) was PCR-cloned into the pMBP-Parallel1 expression vector[41], which expresses the tag-free N-terminal extension region of human TRS (residues 30−74). For ITC experiments, the gene encoding the N-terminal extension region of human TRS (residues 1−74) was sub-cloned into the pET22b(+) vector (Invitrogen) containing a C-terminal hexahistidine (His)-tag, and the gene for 4EHP (residues 45−234) was sub-cloned into the pGST-Parallel1 expression vector[41]. For in vitro binding assays, fly TRS (residues 1−50), zebrafish TRS (residues 1−69), and human full-length TRS were cloned into pET22b(+). Full-length fly, zebrafish, and human 4EHPs and TRS were cloned into pGST-Parallel1[41]. Human 4EHP (residues 45−245) was cloned into pGST-parallel1, and human eIF4A and PABP were cloned into pHis-GST-parallel1. All constructs were confirmed by DNA sequencing. Primer sequences are listed in Supplementary Table 2.

**Antibodies and reagents**. Antibodies were obtained from the following sources: HA (sc-7392, 1:1000), c-Myc (sc-40, 1:1000), GFP (sc-9996, 1:1000), TRS (sc-166146, 1:1000), AlaRS (sc-81712, 1:1000), IRS (sc-271826, 1:1000), PRS (sc-393505, 1:1000), VEGF-A (sc-152, 1:1000), and ANG1 (sc-74528, 1:1000) from Santa Cruz Biotechnology; GST (#2624, 1:2000), eIF4A (#2013, 1:2000), and eIF4G1 (#8701, 1:2000) from Cell Signaling Technology; α-tubulin (T6074, 1:10,000) and FLAG (F3165, 1:10,000) from Sigma-Aldrich; 4EHP (GTX103977, 1:1000) and eIF3L (GTX120119, 1:1000) from Genetex; Strep (#2–1509–001, 1:1000) from IBA; eIF4E (#610269, 1:1000) from BD Biosciences; Alexa Fluor 488 phalloidin (A12379, 1:1000), Alexa 647 (A27040, 1:1000), Alexa 594 (A21201, 1:1000), and V5 (R961–25, 1:1000), HRP-labeled anti-mouse (#31430, 1:20,000), and anti-rabbit (#31460, 1:20,000) secondary antibodies from Thermo Fisher; eIF3A (NBP1–18891, 1:1000), eIF3K (NB100–93304, 1:1000), and eIF3M (NBP1–56654, 1:1000) from Novus; KRS (ab31532, 1:1000) and PABP (ab21060, 1:1000) from Abcam; HIF1-alpha (A300–286A, 1:1000), eIF3B (A301–761A, 1:1000), eIF3C (A300–377A, 1:1000), eIF3D (A301–758A, 1:1000), eIF3E (A302–985A, 1:1000), eIF3F (A303–005A, 1:1000), eIF3G (A301–757A, 1:1000), and eIF3H (A301–754A, 1:1000) from Bethyl; eIF3I (#646701, 1:1000) and eIF3J (#638401, 1:1000) from Biolegend. Transfection was performed using Lipofectamine 2000 (Invitrogen), Fugene HD (Roche), and TurboFect (Thermo Fisher). Strep-Tactin-coated magnetic beads (Magstrep type3 XT beads) were purchased from IBA.

**Liquid chromatography-mass spectrometry (LC-MS) analysis**. HEK293T cells were transfected with pEXPR-IBA105 (empty vector, EV; negative control) or TRS-harboring pEXPR-IBA105 (bait) and lysed in lysis buffer containing 1% NP-40, 20 mM Tris-HCl (pH 8.0), 150 mM NaCl, 2 mM EDTA, 10% glycerol and protease inhibitor cocktail, and lysates were centrifuged at 16,000 g for 20 min at 4 ℃. Cell lysates containing Strep-tagged TRS were then applied to a Strep-Tactin Superflow column (IBA) for affinity purification and washed with Strep wash buffer (100 mM HEPES, pH 8.0, 150 mM NaCl, and 1 mM EDTA), and proteins were eluted with Strep elution buffer (100 mM HEPES, pH 8.0, 150 mM NaCl, 1 mM EDTA, and 2.5 mM desthiobiotin). Eluted proteins were separated by sodium dodecyl sulfate-polyacrylamide gel electrophoresis (SDS-PAGE) and excised into 10 gel pieces. Individual gel pieces were destained and subjected to in-gel digestion using trypsin, and digested peptides were analyzed using an LTQ-Orbitrap Velos (Thermo Fisher) connected to an Easy-nano LC II system (Thermo Fisher) incorporating an autosampler. Dried peptide samples were re-suspended in 70 μL of 0.1% formic acid, and an aliquot (7 μL) was injected onto a reversed-phase peptide trap EASY-Column (length = 2 cm, internal diameter = 100 μM, 5 μM, 120 Å, ReproSil-Pur C18-AQ, Thermo Fisher) and a reversed-phase analytical EASY-Column (length

= 10 cm, internal diameter = 75 μM, 3 μM, 120 Å, ReproSil-Pur C18-AQ, Thermo Fisher). Electrospray ionization (ESI) was subsequently performed using a 30 μM (internal diameter) nano-bore stainless steel online emitter (Thermo Fisher). The duration of the LC gradient was 120 min. Peptides were eluted using a linear gradient of 10−40% buffer B over 98 min (buffer A = 0.1% formic acid in H$_2$O, buffer B = 0.1% formic acid in acetonitrile) at a flow rate of 300 nL min$^{-1}$. The LTQ-Orbitrap Velos mass analyzer was operated in positive ESI mode using collision-induced dissociation (CID) to fragment peptides following separation by HPLC. The temperature and voltage applied to the capillary were 275 °C and 1.9 V, respectively. All data were acquired with the mass spectrometer operating in automatic data-dependent switching mode. MS spectra were scanned from 350 to 2000 m z$^{-1}$ with a resolution of 100,000. The automatic gain control (AGC) target was set at 1,000,000 ions with a maximum fill time of 500 ms. A total of 20 data-dependent MS/MS scans were selected and fragmented in the ion trap using an isolation window of 2.0 m z$^{-1}$, an AGC target value of 10,000 ions, a maximum fill time of 100 ms, a normalized collision energy of 35, and an activation time of 10 ms. Dynamic exclusion was performed with a repeat count of 1, an exclusion duration of 180 s, and a dynamic exclusion list size of 500. The minimum MS ion count for triggering MS/MS was set at 5000 counts. Each sample was analyzed using three LC-MS/MS runs (triplicate runs per sample).

**Interactome data analysis.** Data were integrated using Sage-N Sorcerer 2 software (version 3.5) and used for comprehensive protein identification and characterization. All MS/MS samples were analyzed using Sequest (XCorr Only; Thermo Fisher; version v.27, rev. 11) and X! Tandem (thegpm.org; version CYCLONE 2010.12.01.1) using the UniProt human database (release 2014) containing 162,717 entries. Search parameters were set as follows: cleavage site = full digestion using trypsin/Lys-C (after KR/-) with up to two missed cleavages; precursor and fragment mass tolerance were 25 ppm and 1.0 Da, respectively; carbamidomethylation (+57.021 Da) of cysteine (C) was set in Sequest (XCorr Only) and X! Tandem as a fixed modification; oxidation (+15.995 Da) of methionine (M) was set in Sequest (XCorr Only) and X! Tandem as a variable modification; additional variable modifications Glu- > pyro-Glu (−18.01), ammonia-loss (−17.03), and Gln- > pyro-Glu (−17.03) were searched; and processed data were subsequently transformed into a *.sf file using Scaffold 4 Q + S version 4.6.1 (Proteome Software Inc.). This program was also used to validate MS/MS-based peptide and protein identifications and to perform quantitative analysis. Peptide identification was accepted if the probability was > 90.0% according to the Peptide Prophet algorithm[42] with Scaffold delta-mass correction, and if the probability was > 90.0% for at least one identified peptide. Protein probabilities were assigned by the Protein Prophet algorithm[43]. Proteins containing similar peptides that could not be differentiated based on MS/MS analysis alone were grouped to satisfy the principles of parsimony. SAINT analysis was performed to assign a confidence score to binding partners of negative control and bait sample proteins, and a SAINT score ≥ 0.9 was accepted for prey proteins[44].

**Yeast two-hybrid analysis.** cDNA encoding human TRS was obtained by PCR using specific primers and ligated into the pEG202 vector[45] (for the construction of the LexA fusion protein). The LexA-human TRS fusion protein was used as bait to screen binding proteins from the HeLa cell cDNA library in which proteins were expressed as B42 fusion proteins[46].

**Preparation of cell lysates and immunoprecipitation.** Cells were dissolved in lysis buffer containing 1% Triton X-100, 50 mM HEPES (pH 7.4), 150 mM NaCl, 2 mM EDTA, 10 mM pyrophosphate, 10 mM glycerophosphate and protease inhibitor cocktail (Calbiochem), and lysates were centrifuged at 16,000 g for 15 min. Extracted proteins (20 μg) were fractionated by SDS-PAGE. For Strep-tagged protein precipitation, cells were lysed and Magstrep type3 XT beads were added and used according to the manufacturer's instructions. For immunoprecipitation, cells were lysed and primary antibodies were added and incubated with agitation for 4 h at 4 °C. A 50% slurry of Protein G Agarose (Invitrogen) was added, and incubation continued for 4 h. After washing three times with ice-cold lysis buffer, precipitates were dissolved in SDS sample buffer and separated by SDS-PAGE.

**Bimolecular fluorescence complementation (BiFC) assay.** AlaRS, EPRS, KRS, TRS and WRS were cloned into pBiFC-VN173 (FLAG tag). 4EHP, eIF4E1, and eIF4E3 were cloned into pBiFC-VC155 (HA tag). CHO cells were co-transfected with pBiFC-VN173-ARSs together with pBiFC-VC155-eIF4E isoforms. Cells were fixed with 100% methanol for 7 min and incubated with blocking solution (3% CAS) for 15 min at room temperature. After blocking, cells were stained with HA, Alexa 647, FLAG, and Alexa 594 for 1 h at room temperature. DAPI was used to stain nuclei. Cells were washed three times with PBS after every step. After mounting, the TRS-4EHP interaction was observed by fluorescence and confocal microscopy (Nikon, A1Rsi).

**Protein expression and purification.** Expression of MBP-His-tagged 4EHP (residues 45−234) in complex with the N-terminal extension domain of TRS (residues 30−74) was induced in *E. coli* BL21 (DE3) cells by treatment with 0.5 mM IPTG for 16 h at 18 °C. TRS-His or His-GST-tagged eIF4A were overexpressed

in *E. coli* BL21-CodonPlus (DE3)-RIPL cells by induction with 0.5 mM IPTG for 16 h at 18 °C. Recombinant proteins were purified using Ni-NTA affinity chromatography, treated with recombinant tobacco etch virus (rTEV) protease (GIBCO) to remove MBP-His- or His-GST-tags, then purified by size exclusion chromatography and additional Ni-NTA affinity chromatography. Expression of N-terminal GST-fused human 4EHP (residues 45−245) was induced in *E. coli* BL21 Star (DE3) cells with 0.5 mM IPTG for 16 h at 18 °C. His-GST-tagged PABP was overexpressed in *E. coli* BL21-CodonPlus (DE3)-RIPL cells by induction with 0.5 mM IPTG for 16 h at 18 °C. Recombinant proteins were purified affinity column chromatography with immobilized glutathione resin. The GST-tag was removed from 4EHP by on-resin rTEV protease treatment for 16 h at 4 °C. Eluted proteins were further purified by size exclusion chromatography, and peak fractions containing recombinant proteins were concentrated for further study. HEK293T cells grown in 150 mm culture dishes were transfected with the TRS-Strep construct using X-tremeGENE HP transfection reagent (Roche). After 24 h, harvested cells were lysed using Strep lysis buffer (100 mM Tris-HCl, pH 8.0, 150 mM NaCl, 5 mM EDTA, protease inhibitor cocktail, and phosphatase inhibitor cocktail) and clarified by centrifugation at 16,000 g for 20 min at 4 °C. Cell lysates were incubated with Strep-Tactin resin (IBA) for 2 h with agitation, applied to the column for gravity elution, and washed with Strep wash buffer (100 mM HEPES, pH 8.0, 150 mM NaCl, and 1 mM EDTA). Bound proteins were eluted with Strep elution buffer (100 mM HEPES, pH 8.0, 150 mM NaCl, 1 mM EDTA, and 2.5 mM desthiobiotin).

**In vitro binding assay.** The protein mixture containing TRS-Strep and 4EHP (residues 45−245) at a 1:3 molar ratio was incubated for 8 h at 4 °C. TRS-Strep and His-GST-tagged PABP were mixed in a 1:2 molar ratio and incubated for 8 h at 4 °C. The mixture was loaded onto Strep-Tactin resin for 30 min at 4 °C, thoroughly washed five times with ice-cold wash buffer (50 mM Tris-HCl, pH 7.5, 300 mM NaCl, 1 mM EDTA, and 2% NP-40), and eluted with Strep elution buffer (100 mM HEPES, pH 8.0, 150 mM NaCl, and 1 mM EDTA supplemented with 2.5 mM desthiobiotin) for further analysis. Purified TRS-Strep and eIF4A were mixed in a molar ratio of 1:2 and incubated with eIF4A antibody (1:200; Cell Signaling Technology) for 4 h at 4 °C. The mixture was incubated with 30 μl of Protein A/G PLUS-agarose beads (Santa Cruz Biotechnology) for 4 h at 4 °C and thoroughly washed five times with ice-cold wash buffer described above before immunoblot analysis. Plasmids expressing His-tagged fly TRS (residues 1−50), zebrafish TRS (residues 1−69), and human TRS (residues 1−74) were transformed into *E. coli* BL21 (DE3), *E. coli* Rosetta-gami (DE3) pLysS, and *E. coli* BL21 (DE3), respectively. Plasmids expressing GST-fused fly, zebrafish, and human 4EHP were transformed into *E. coli* Rosetta-gami (DE3) pLysS, *E. coli* C43 (DE3), and *E. coli* BLR (DE3), respectively. Protein expression was induced with 0.5 mM IPTG at 18 °C for 20 h. Harvested cells were re-suspended in Buffer A (50 mM Tris-HCl, pH 8.0, 300 mM NaCl, and 10 mM imidazole), lysed by sonication on ice, and centrifuged at 12,000 g at 4 °C for 1 h. Supernatants containing His-tagged TRS and GST-fused 4EHP pairs from different species were loaded onto an Ni-NTA agarose column, washed extensively with Buffer A, and eluted with 250 mM imidazole. All eluted proteins were analyzed by SDS-PAGE and stained with Coomassie Brilliant Blue.

**In vitro pull-down assay of GST-TRS and FLAG-tagged proteins.** Plasmid encoding GST-fused TRS was transformed into *E. coli* BL21 (DE3) cells, and protein expression was induced with 0.5 mM IPTG at 18 °C for 16 h. Harvested cells were re-suspended in ice-cold Buffer B (PBS containing 0.2% NP-40, 5% glycerol, 1 mM EDTA and 1 mM DTT), lysed by sonication on ice, and centrifuged at 12,000 g at 4 °C for 1 h. Supernatant containing GST-fused TRS was incubated with glutathione-Sepharose 4B (GE Healthcare) at 4 °C for 4 h, and thoroughly washed three times with ice-cold Buffer B. For the pull-down assay, 293T cells transfected with FLAG-tagged proteins were lysed, mixed with purified GST-TRS, and incubated at 4 °C for 4 h with gentle agitation. After thorough washing three times with ice-cold Buffer B, precipitates were dissolved in SDS sample buffer and separated by SDS-PAGE.

**Isothermal titration calorimetry (ITC).** Purified 4EHP (residues 45−234) and C-terminally His-tagged TRS (residues 1−74) proteins were degassed by vacuum aspiration for 15 min before loading, and titration was performed at 25 °C with a VP-ITC titration calorimeter (MicroCal Inc.). TRS (residues 1−74, 0.6 mM) was placed in the syringe and titrated against 4EHP (residues 45−234, 0.035 mM). Raw data were fitted to a single binding site model using Origin version 7.0 supplied with the instrument.

**Crystallization and structure determination.** Crystals of 4EHP (residues 45−234) in complex with the TRS N-terminal domain (residues 1−74) obtained using in situ proteolysis[47] diffracted well, but phases could not be obtained. Therefore, several truncated variants of the N-terminal extension domain of TRS in complex with 4EHP (residues 45−234) were purified as described above and crystallized using the sitting-drop vapor-diffusion method at 21 °C. Initial crystals of the 4EHP (residues 45−234)-TRS (residues 30−74) complex were optimized, and the best crystals were grown in 3 days in 20% polyethylene glycol (PEG) 8000

and 0.1 M CHES pH 9.5. Crystals were transferred into reservoir buffer containing 20% (v v⁻¹) glycerol, mounted immediately in a -173 °C nitrogen gas stream, and diffraction data were collected at 1.9 Å resolution and processed with the HKL2000 package[48]. The structure was solved by molecular replacement using Phaser-MR in Phenix[49] with the 4EHP structure (PDB 2JGB) as a search model. Model building and refinement were performed using COOT[50] and Phenix.refine[49]. The final model includes residues Lys45 to Asp219 of 4EHP and Pro49 to Glu74 of TRS UNE-T. Residues Pro69−Tyr78 and Ser220−Val234 of 4EHP and Gly30−Asn48 of TRS were not included in the final model because they were not observed in the electron density map, presumably due to high flexibility. Crystallographic data are summarized in Supplementary Table 1.

**Mutagenesis**. Mutations in *TRS*, *4EHP*, and 5′ UTR-167 were generated by site-directed mutagenesis using QuickChange (Agilent) following the manufacturer's instructions, and confirmed by DNA sequencing. Primer sequences are listed in Supplementary Table 2.

**Aminoacylation assay**. The aminoacylation assay was carried out in buffer containing 4 mM DTT, 50 mM HEPES-KOH pH 7.6, 20 mM KCl, 10 mM MgCl₂, 5 mM ATP, 2 mg mL⁻¹ yeast tRNA (Roche), various concentrations of [³H] Thr (American Radiolabeled Chemicals), and 100 nM TRS. Reactions were initiated with enzyme and conducted in a 37 °C heat block. Aliquots (10 μL) were taken at different time points and quenched on Whatman filter pads presoaked with 5% trichloroacetic acid (TCA). Pads were washed three times for 10 min each time with cold 5% TCA and once with cold 100% ethanol, and dried, and radioactivity was quantified using a scintillation counter (Beckman Coulter).

**Cap-binding assay**. Cells were washed with PBS and lysed in lysis buffer (20 mM Tris-HCl pH 7.5, 100 mM NaCl, 25 mM MgCl₂, 0.5% NP-40, and protease inhibitor cocktail). Extracts were clarified by centrifugation at 3000 g for 10 min at 4 °C. Supernatants were precleared with 30 μL of agarose beads (Sigma) for 1 h at 4 °C, beads were removed by centrifugation at 500 g for 1 min, and supernatants were incubated with 50 μL of m⁷GTP-Sepharose beads (Jena Bioscience) for 2 h at 4 °C. Pelleted beads were washed four times with 0.5 mL of lysis buffer, re-suspended in sample buffer, and boiled for 5 min. m⁷GTP-bound proteins, as well as 5% of the initial sample taken just before the m⁷GTP beads were added, were subjected to SDS-PAGE.

**RNA interference**. Cells were transfected with duplex siRNA using Lipofectamine 2000 (Invitrogen) according to the manufacturer's instructions. ON-TARGETplus SMARTpool siRNAs against TRS, AlaRS, 4EHP, and eIF4E were purchased from GE Healthcare Dharmacon. Stealth RNAi siRNAs against eIF4G were purchased from Thermo Fisher. A non-targeting siRNA was used as a control. Cells were incubated with siRNAs for 36−72 h. Sequences of siRNAs are listed in Supplementary Table 2.

**Puromycin incorporation assay (SUnSET Assay)**. Cells were incubated with 1 μM puromycin (Thermo Fisher) for 30 min, washed with ice-cold PBS, lysed with lysis buffer, separated by SDS-PAGE, and immunoblotted with mouse anti-puromycin monoclonal antibody (Millipore). Ponceau S stain (INtRON) was used for normalization.

**[³⁵S] Met incorporation assay**. Cells were incubated in Met-free media containing 10 μCi mL⁻¹ [³⁵S] Met (American Radiolabeled Chemicals) for 1 h and washed twice with ice-cold PBS. After a 30 min treatment with 5% TCA, cells were washed twice with ice-cold PBS and solubilized in 0.5 N NaOH. An aliquot was analyzed using a liquid scintillation counter (Beckman Coulter).

**RNA immunoprecipitation and sequencing (RIP-seq)**. RNA immunoprecipitation was performed using a Dynabeads Co-Immunoprecipitation Kit (Thermo Fisher) according to the manufacturer's protocol. In brief, lysed HEK293 cells (5 × 10⁸ cells per sample) were incubated with mouse IgG, anti-TRS, anti-IRS, anti-PRS, or anti-AlaRS antibody-coupled Dynabeads for 45 min at 4 °C. After multiple washes, RNA was extracted by vortexing for 30 s with phenol:chloroform:isoamyl alcohol (PCI) buffer (pH 4.5) and centrifuged at 16,000 g for 2 min. The upper phase was transferred to a fresh tube, and RNA was precipitated by adding 20 mg mL⁻¹ glycogen (Invitrogen), a 0.1 volume of 3 M sodium acetate (pH 5.5), and a 2.5 volume of 100% ethanol. Mixtures were incubated at -20 °C for 1 h and centrifuged at 16,000 × g for 5 min, and RNA pellets were air-dried and re-suspended in RNase-free water for sequencing. The sequencing library was prepared using the TruSeq RNA sample preparation kit v2 (Illumina)[51]. In brief, mRNA derived from total RNA using poly-T oligo-attached magnetic beads was fragmented and converted into cDNA, adapters were ligated, and fragments were amplified by PCR. Paired-end sequencing (101 × 2) was performed using a Hiseq-2000 system (Illumina). Reference genome sequence data from *Homo sapiens* were obtained from the University of California Santa Cruz Genome Browser Gateway (assembly ID: hg19). The reference genome index was built using SAMtools (v. 0.1.19) and the Bowtie2-build component of Bowtie2 (v. 2.1.0). Reads were mapped to the reference genome using Tophat2 (v. 2.0). The number of reads per kilobase per million mapped reads (rpkm) for each gene of the 46,895 RefCSeq (UCSC hg19) gene models was calculated using Cufflinks (v. 2.2.1). All statistical analyses were performed using R (v. 3.1.0). For functional annotation, DAVID (Database for Annotation, Visualization and Integrated Discovery; http://david.abcc.ncifcrf.gov) was used[52], and the enriched gene set was obtained from Ensemble Biomart (http://www.biomart.org)[53].

**RNA immunoprecipitation and reverse transcription-PCR**. Cells were cross-linked with 1% formaldehyde for 10 min at room temperature, and crosslinking was quenched by adding 20 mM glycine for 5 min followed by two washes with cold PBS. Cells were lysed in 1 mL of RIPA buffer containing protease and RNase inhibitors. Samples were sonicated at 20% amplitude for two cycles of 30 s (2 s on / 2 s off) with a 1 min pause between cycles. Samples were centrifuged at 16,000 × g for 3 min at 4 °C, and 50 μL of the supernatant was retained as an INPUT control. Subsequently, samples were precleared using Dynabeads for 15 min at 4 °C. Immunoprecipitation was carried out using rabbit anti-TRS and rabbit IgG overnight at 4 °C, and samples were incubated with Dynabeads for 1 h at 4 °C. Beads were washed five times with RIPA buffer and collected using a magnetic stand (Millipore). Crosslinking was reversed by incubating for 1 h at 65 °C. RNA extraction was performed using TRIzol reagent (Invitrogen) according to the manufacturer's instructions to identify interacting RNA segments. Primer sequences are listed in Supplementary Table 2.

**RNA secondary structure prediction**. The human VEGF 5′ UTR sequence (GenBank accession no. NM_001025366) was submitted to the RNA fold web server (http://rna.tbi.univie.ac.at/cgi-bin/RNAfold.cgi)[54] for secondary structure prediction.

**Luciferase reporter assay**. Luciferase assays were performed using the Luciferase and *Renilla* Luciferase Assay Systems (Promega) according to the manufacturer's instructions. Briefly, HEK293T cells were transfected with the luciferase reporter plasmid pGL2 combined with the VEGF 5′ UTR sequence. pRL *Renilla* luciferase (Promega) was co-transfected and used for normalization of transfection efficiency. At 24 h after transfection, cells were lysed and reporter activity was analyzed.

**Real-time RT-PCR**. Total RNA was extracted from cells using the RNeasy RNA extraction Mini-Kit (QIAGEN). Purified RNA was treated with RNase-free DNase at 37 °C for 30 min. Quantitative PCR was performed using gene-specific primer sets and SYBR Green Supermix (BioRAD). Real-time PCR was carried out using a LightCycler 96 (Roche Diagnostics) according to the manufacturer's instructions. Data were normalized against glyceraldehyde-3-phosphate dehydrogenase expression, and relative expression was calculated using the ΔΔCT method. Primer sequences are listed in Supplementary Table 2.

**Enzyme-linked immunosorbent assay (ELISA)**. The angiogenic cytokine concentration in cell culture supernatants was measured by ELISA according to the manufacturer's instructions using VEGF and angiogenin kits (R&D Systems).

**Secretion assay**. For cell culture media samples, cells were incubated for 16 h in serum-free medium and centrifuged for 5 min at 200 g, and supernatants were concentrated 10-fold using an Amicon Ultra-4 centrifugal filter (Millipore).

**In vitro tube formation assay**. HUVECs were cultured until 95−100% confluent, seeded in 48-well plates coated with Matrigel Basement Membrane Matrix GFR (BD Biosciences), and incubated at 37 °C in EGM-2 media with 2% FBS and various media for 4−8 h. Cells were fixed with 4% paraformaldehyde, stained with Alexa Fluor 488 phalloidin (Life Technologies), and imaged by fluorescence microscopy. Tube structures were quantified using ImageJ (NIH).

**Cell migration assay**. Cell migration was determined using 24-well Trans-well chambers with a polycarbonate membrane (8 μM pore size; Corning). HUVECs were suspended in EBM2 basal medium and added to the top chamber at a density of 1 × 10⁴ cells per well. To determine the effect of siTRS and si4EHP, various media were placed in the bottom chamber, and cells were allowed to migrate for 6 h at 37 °C, and were fixed with 70% methanol in PBS for 30 min, washed with PBS three times, stained with hematoxylin (Sigma-Aldrich) for 10 min, and washed with distilled water. Non-migrating cells were removed from the top face of the membrane with a cotton swab, and membranes were excised from the chamber and mounted with Gel Mount (Biomeda). Migrating cells attached to the bottom face of the membrane were then counted using three randomly selected views in high-power fields (×20).

**Zebrafish husbandry and embryo fixation**. The endothelial-specific transgenic line *Tg(kdrl:EGFP)* and WT AB line were maintained at the zebrafish facility in the Korea Research Institute of Bioscience and Biotechnology (KRIBB) under a 14:10 h light:dark cycle. Embryos were collected and raised at 28.5 °C, treated

with 1-phenyl 2-thiourea (PTU, Sigma) to prevent pigmentation, and fixed at 52 h post-fertilization (hpf) with 4% paraformaldehyde in PBS and staining solution (0.15 mM $CaCl_2$ and 4% sucrose in $1 \times$ PBS) to preserve EGFP fluorescence after fixation. Fixed zebrafish larvae were embedded on 1.3% low melting point agarose (Promega), and confocal images were taken using an Olympus FV1000 confocal microscope. Z-projected images were used to measure the length and branching points of vessels using ImageJ. Zebrafish husbandry and animal care were carried out in accordance with the guidelines of KRIBB, and experimental protocols were approved by KRIBB-IACUC (approval number: KRIBB-AEC-17117).

**Morpholinos (MOs) and mRNA injection into zebrafish embryos.** Splice-blocking MOs for *trs* (Accession: NM_001122786.1 GI: 218563695) targeting exon7 and *4ehp* (Accession: NM_001014815.2 GI: 212549733) targeting exon3 were constructed using i6e7 MO (5′-ACTAGAGGAAAGAGAGAGCGCAGATT-3′) and 4ehp MO (5′-GCGTGTGTGTAGGTTACCGAAGCAA-3′) from GeneTools (http://www.gene-tools.com/), respectively. One nl of *trs* i6e7 MO (200 μM) or *4ehp* MO (200 μM) was injected to transiently knock-down TRS and 4EHP expression without causing gross morphological defects to developing larvae. The same dose of a standard control MO (5′-CCTCTTACCTCAGTTACAATTTATA-3′) was used as a negative control. Primers used for validating MO knock-down efficiency were as follows: F primer (5′-TGTTACGGACCACCCATCGA-3′) and R primer (5′-CCCTCCCAGTATGTAGAG-3′) for *trs* i6e7 MO; F primer (5′-GCGG ATCCAGACACTCTTCT-3′) and R primer (5′-CACATGGGTTTGATTCCTT CCT-3′) for *4ehp* MO. Functional mRNAs were *in vitro* synthesized using the mMESSAGE mMACHINE SP6 Transcription Kit (Ambion, AM1340) according to the manual, with *Not*I-linearized plasmid containing an appropriate full-length cDNA (TRS WT, TRS I55D mutant, 4EHP WT or mCherry as a negative control) in the pCS2 + vector. In rescue experiments, 0.4 ng of each mRNA was co-injected with *trs* i6e7 MO or *4ehp* MO. All injections were carried out when fertilized eggs were at the 1–2 cell stage.

**Whole-mount RNA *in situ* hybridization (WISH).** Digoxigenin (DIG)-labeled anti-sense RNA probes for WISH were prepared using AB WT cDNA as a template to amply 1 kb fragments for each probe (F primer 5′-ACAGTACATTGATGAG CGCC-3′ and R primer 5′-TAGTGCTGCCAGTGTCCTGA-3′ for *trs*; F primer 5′-TTTGACGCCCTGAAAGATGA-3′ and R primer 5′-CCTCCACTTGCGTTCAC TAG-3′ for *4ehp*). PCR products were cloned using a Zero Blunt TOPO PCR Cloning Kit (Invitrogen, 450245). Cloned DNA was linearized by *Xho*I and *Hind*III restriction enzymes, and *4ehp* and *trs* anti-sense DIG-labeled RNA probes were synthesized by SP6 RNA polymerase (Roche, 10810274001) and T7 RNA polymerase (Roche, 10881767001), respectively, using a DIG RNA labeling kit (Roche, 11277073910). The WISH procedure was performed following the method described by Thisse and Thisse[55]. Briefly, 4% PFA-fixed embryos were washed with $1 \times$ PBST ($1 \times$ PBS + 0.1% Tween 20 in diethyl pyrocarbonate water), serially transferred through 25%, 50%, 75%, and 100% methanol in $1 \times$ PBST, and stored at −20 °C until needed. On the 1st day, embryos were washed and permeabilized by treating with proteinase K (Sigma, P2308) at appropriate concentrations for different developmental stages of embryos. Post-fixed embryos were washed and hybridized in Hyb + solution with the probes overnight at 70 °C. On the 2nd day, after serial washing with $2 \times$ SSCTw/50% formamide, $2 \times$ SSCT, $0.2 \times$ SSCT at 70 °C, and $1 \times$ PBST at room temperature, samples were blocked with 5% horse serum (Sigma, H1138) in $1 \times$ PBST and incubated with anti-DIG-AP Fab fragments (Roche, 11093274910) overnight at 4 °C. On the 3rd day, samples pre-incubated in staining buffer were incubated with BCIP/NTP (Roche, 11681451001) substrate solution for color reaction. Finally, samples were washed with $1 \times$ PBST, and then replaced with 25%, 50%, 75%, and 100% glycerol for imaging.

**Preparation of total protein from mouse tissues.** Embryos were collected from C57BL/6 mice at post-coital days 8.5−12.5. For detection of proteins, embryos were homogenized in 1% Triton X-100, 50 mM HEPES pH 7.4, 150 mM NaCl, 2 mM EDTA, 10 mM pyrophosphate, 10 mM glycerophosphate, and protease inhibitor cocktail at 4 °C for 30 s, incubated on ice for 30 s, and homogenized again for 30 s using a homogenizer microtube (COSMOBIO). Lysates were centrifuged at 16,000 g for 15 min at 4 °C, and supernatants were collected.

**Statistical analyses.** Statistical analyses were performed using Prism 6 (Graph-Pad). Bar graphs were plotted as means ± standard deviation, and statistical significance was denoted as $*p < 0.05$, $**p < 0.01$, $***p < 0.001$, $****p < 0.0001$, or not significant (NS) vs. the control group. Bar graphs in zebrafish experiments were plotted as means ± standard deviation, and statistical significance was denoted by one-way analysis of variance (ANOVA) and denoted as $*p < 0.05$, $**p < 0.01$, $***p < 0.001$, $****p < 0.0001$, or not significant (NS) vs. the control group.

## Data availability
All data supporting the findings of this study are available within the article and its supplementary information files, or from the corresponding author upon request. Coordinates and other structure-related information have been deposited in the Protein Data Bank (PDB) under PDB code 5XLN. RNA immunoprecipitation and sequencing

(RIP-seq) data have been deposited in the Gene Expression Omnibus (GEO) database under accession code GSE120182. Source data for Figs. 1 to 7 and Supplementary Figures 1 to 13 are provided as a Source Data file.

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

## Acknowledgements

We thank beamline staff at the Photon Factory, Japan (BL-5A, BL-17A) and Pohang Light Source, Korea (BL-5C) for assistance during X-ray diffraction experiments, and Dr. Jong Hwan Kim at KRIBB for initial processing of mRNA sequencing data. This study was supported by the National Research Foundation of Korea, funded by the Ministry of Science and ICT of Korea (NRF-M3A6A4–2010–0029785 to S.K.; NRF-2010–0029767 to M.H.K.), and the KRIBB Initiative Program (M.H.K.).

## Author contributions

S.J.J., S.P., L.T.N., M.H.K., and S.K. designed experiments. S.J.J. conducted most of the cell biology experiments, including RIP-seq analysis, with help from S.P., I.Y., E.-Y.L., J.-H.L., J.H.K., H.K.K., D.K., W.S.Y., and S.-Y.K. S.P., L.T.N., and J.H. performed structural and biochemical experiments with help from C.Y.L. H.-K.G., J.-S.L., and K.Y. carried out zebrafish experiments. S.J.J., N.S., M.H.K., and S.K. analyzed and interpreted the results and wrote the manuscript with input from all other authors.

## Additional information

**Competing interests:** The authors declare no competing interests.

**Journal Peer Review Information:** *Nature Communications* thanks the anonymous reviewer(s) for their contribution to the peer review of this work. Peer reviewer reports are available.

