## [Peer Review File · Nature Communications]

Reviewers' comments:

Reviewer #1 (Remarks to the Author):

This manuscript represents a paradigm shift in our understanding of eukaryotic translation and when acceptable is clearly suited for Nature. As may be anticipated with a new idea, there are a variety of concerns given the different view point. Studies on the mechanism of eukaryotic initiation began with individual components and then the establishment of a pathway followed by in vivo and genetic experiments. These studies seem to have started with the big picture and worked towards some of the smaller details.

At the very outset, if one is to propose that TRS, eIF4E2 and eIF4A are to form a functional complex that replaces eIF4F, where is the evidence (i.e. mix the proteins together and show they all elute as a high molecular weight peak; MW about 200,000)? Secondly, the enzymatic properties of eIF4F are quite different from those for eIF4A; Is this also true for the authors' new complex?

Much of the authors' data reflect various pulldown experiments followed by Western blotting. While specificity may be seen in that eIF4A2 is pulled down (Figure 1A) and eIF4A1 is not, it is not clear how many other interacting proteins might have been missed because they were not blotted for. That is, if the pulldown was subjected to SDS gel electrophoresis, how many protein bands would be visualized? Secondly, it is not clear from either the figure legends or methods as to how often the experiment was performed nor are error bars evident in many of the figures (i.e. what is the variation between blots and how are they normalized?).

Specific concerns

1 The TRS is a dimer. How representative are the results achieved with a monomer (mostly for pieces of TRS such as VN, VC, UNE, etc. as either pull downs or crystal structure)?

2 There does not seem to be a mathematical representation of the strength/relative affinity of TRS for other proteins. It would be of value to have some better idea as to what this might be as this is not all that evident from supplementary figure 1.

3 The authors often describe pull down results as indicating "a specific interaction" if one or two other proteins are not seen. This is insufficient (see above).

4 The TRS-4EHP interaction is cited as specific for vertebrates, yet there is only a single non-vertebrate tested (fly). It would be useful to extend this to at least two other negative species.

5 Using AlaRS as a control, the RNA pulldown with TRS was reduced from 7,255 (IgG control) to 2,928. How much more of a reduction might be seen if additionally ProRS and IleRS RNA pulldowns were done (i.e. other tRNA synthetases that have either A or C as the first or second nucleotide in their codon)?

6 The use of either "TRS expressing cells" or siTRS transfected cells seems problematic. First, there is no quantitation as to the level of overexpression of TRS and secondly, the affect observed is barely two-fold over empty vector control. With respect to the use of siRNAs, it is anticipated that protein synthesis requires all 20 synthetases. As the measurement of luciferase is 36 to 72 hours after siRNA treatment, it is not clear what level of protein synthesis is actually occurring. Is there the expected decrease in protein synthesis (as might be measured by S35 methionine incorporation)?

7 Given the apparent effectiveness of the TRS knockdown (supplementary figure 4A), it is surprising that the level of VEGF expression is only reduced 30 to 40% in HEK293T cells (figure 5A).

8 In supplementary figure 7 it is shown that m7GTP-Sepharosecan pulldown TRS (via its interaction with 4EHP). What other proteins are also pulled down with 4EHP under these conditions?

9 Figure 7F is supposed to show an interaction with eIF3 and TRS. However, only a few eIF3 subunits appear visible. Why are not all of the eIF3 subunits pull down?

Reviewer #2 (Remarks to the Author):

In this manuscript Jeong et al. have shown that human TRS (hTRS) has acquired additional functionality of regulating translation of other genes involved in a variety of physiological processes, the majority being related to vertebrate system development. Using pull-down and co-IP assays extensively, they have identified different domains/regions of hTRS that interact with specific set of mRNAs (e.g., VEGF) and other protein factors (4EHP) to form an initiation complex that is analogous to the canonical eIF4F complex, but functions independently of the latter. The authors have also solved the co-crystal structure of (4EHP + hTRS UNE-T) that has identified the specific interactions between these two proteins, and corroborates other findings. They have further demonstrated that such a regulation of translation by TRS is specific to vertebrate lineage, since fruit fly TRS fails to form such a complex. Their functional assays show the effect of knocking down of TRS–4EHP initiation machinery on target gene expression and its phenotypic manifestations. The authors have performed significant amount of work by approaching the problem from structural, mechanistic and cell biological perspectives. Overall, the current study has identified a hitherto unknown function of hTRS in particular, and vertebrate TRS in general, in the translational regulation of a multitude of metabolic processes, suggesting that such a gain-of-function in vertebrate lineage have played a role in the delineation of vertebrates from invertebrates. The results are novel and provide a totally new dimension to the problem of translation regulation in vertebrates.

Concerns

- Page 10, Supplementary fig 3a: The changes observed in luciferase assay upon changing the UGU (Thr) anticodon-like loop to even AAA (Phe) anticodon-like loop are less than 5%, when compared to luciferase activity in the case of UGU (Thr) anticodon-like loop. This implies that recognition of anticodon-like loop structure by TRS may not be strictly sequence specific. The authors should check the effect of loop proximity to the translation start site (i.e. the position of the loop in 5' UTR) as well as loop length on translation regulation (luciferase expression). Furthermore, they should also check for the presence of such loops in other genes, at least in the case of ANG gene that was tested in the study.

- In vitro pull-down assay need to be checked to confirm direct interaction between TRS and PABP because the latter's detection in co-IP with TRS ABD can also be due to interaction of mRNAs with ABD.

- TRS LLL mutant also can be probed for interaction with eIF4A, PABP and eIF3 to rule out any overlap between binding sites of these factors and that of UGU anticodon-like loop.

- In the model, emphasis should be put on the importance of ABD's recognition of the UGU anticodon-like loop, leading to the binding of other factors to form the initiation complex.

- The authors need to tone down their claim about invertebrate-to-vertebrate transition. I would suggest that this point can be emphasized in 'Discussion' and not in 'Abstract'.

Minor comments

- Page 3, para 2: "Cell condition-specific...eIF4 isomers mediate...sequesters eIF4E." Change "isomers" to "isoforms".
- Page 4, para 2: "The machinery includes...to eF4F composed...and eIF4A." Change "eF4F" to "eIF4F".
- Page 6, para 3: What is meant by "dorsal and lateral surfaces of eIF4E"? It is better to explain these terms (preferably by a supplementary fig, if possible) for the convenience of the general reader.
- Page 6, para 3: "Furthermore,...interacting with eIF4E..." change "interacting" to "interact".
- Pages 6/7, paras 3/1: "However, TRS does not appear...immediately after Asp78 (Fig. 1f,g)." This point is not clear! It is better to elaborate on this point a bit. How does TGS domain beginning immediately after Asp78 not lead to TRS not utilizing the non-canonical motifs and auxiliary sequences? The reference of Fig 1f,g does not seem to support/clarify this point at all. Moreover, it is not clear whether TRS possesses non-canonical motifs and auxiliary sequences.
- Page 8, para 2: "Mutation of ...motif of zebrafish 4EHP..." change "zebrafish 4EHP" to "zebrafish TRS".
- Page 9, para 2: "Remarkably, most genes...development of nervous, skeletal, and circulation systems..." Nervous and circulatory systems are not unique to vertebrates! Thus, the sentence needs to be modified.
- Page 11, para 2: "A slight reduction...was silenced (Supplementary Fig. 4a)." Explanation? Error bars?
- Page 11, para 2: "Silencing of TRS...by oxygen tension (Supplementary Fig. 4e)." What is the relevance of this aspect in the context of the study?
- Page 12, para 1: "Tube formation and...TRS (M60K)-expressing cells (Fig. 5d and Supplementary Fig. 5d)." Enhancement in the case of M60K is seen for both tube length and cell migration, but to a lesser extent than wild-type. Hence, the statement is incorrect and needs to be modified accordingly.
- Page 16, para 2: "During evolution,...complexity of higher organisms." Whether increasing complexity resulted in the acquisition of additional functionalities or the latter resulted in increased complexity? This is a contentious and debatable statement, and therefore needs to be modified accordingly.
- Page 17, para 4, Supplementary fig 9: Since this is a dataset generated in the present work, it should be reported in the 'Results' section, and not merely in the 'Discussion' section.
- Crystallographic table: round off the unit cell dimensions to 2 decimal places; mention the number of unique reflections in the highest resolution shell; mention the resolution range for the highest resolution shell used in refinement; mention the number of reflections used in refinement; mention

avg B-factor for protein, ligand and water separately; change “Geometry (%)” to “Ramachandran statistics (%)”; mention what % of reflections were used for Rfree calculation during refinement.

- Fig 1h: Provide the standard deviation value for Kd.
- Fig 2a: The Fo-Fc map should be an unbiased map generated before modeling of the ligand.
- Supplementary fig 3b: Show quantification of hTRS to show dose-dependent increase in protein level
- Supplementary fig 4c: Why do eIF4E bands appear so faint? Error bars?
- Error bars missing in Supplementary fig 4a,c,e.

Reviewer #3 (Remarks to the Author):

The authors present a study that proposes a new function of an aminoacyl-tRNA synthetase in stimulating cap-dependent translation of VEGF. Overall, this is an interesting study and has the potential to transform our understanding of how selective translation of VEGF can be regulated by 4EHP. A strength of the study is the structural model that reveals how 4EHP binds to TRS. The validation of this interaction using structure guided mutants is impressive (although mutations in zebrafish TRS did not clearly result in loss of binding (Fig. 3d)).

Several weaknesses of the study lowers enthusiasm regarding the validity of the proposed model. Basically, more rigorous experiments are needed to strengthen the findings.

There is a considerable emphasis on interaction models being generated from immunoprecipitation data from the over-expression of epitope tagged proteins in cells. The possibility of indirect binding partners being involved for interactions is not rigorously explored. This is particularly evident in the case of eIF4A and eIF3 binding to TRS. The ability of TRS to bind eIF4A and eIF3 needs to be more rigorously tested using biochemical and biophysical approaches to substantiate the proposed model whereby TRS functions in place of eIF4G.

The VEGF translation reporter assay requires the over-expression of 4EHP and TRS. This raises some concern about how this interaction promotes translation in an endogenous system when components are not over-expressed.

Detailed comments:

As the authors point out, the translation of VEGF can be regulated by hypoxia. It is shown that knockdown of 4EHP still inhibits VEGF translation in reduced oxygen conditions (FIG S4). It should be noted that the authors need to add error bars to all the data presented in Fig. S4 to show that the data are statistically relevant.

The translation of VEGF has been shown to be sensitive to both eIF4E and eIF4G levels, in both normal and hypoxic conditions (Int J Cancer J Int Du Cancer 1996; 65:785-90; PMID:8631593; Mol Cell. 2007;28:501–512). The authors need to confirm that eIF4E and eIF4G knockdown do in fact

reduce VEGF translation in hypoxic conditions in their system.

The claim that eIF4A binds directly to TRS is most unexpected. This finding should be more rigorously investigated to substantiate the proposed model. Is the translation of VEGF sensitive to eIF4A inhibition by PDCD4 and hippuristanol? The in vitro binding assays (ITC and pull-downs using purified recombinant full-length factors) should be repeated with the addition of eIF4A. A mutant TRS that cannot bind to eIF4A should also be generated and tested in the TRS over-expression VEGF reporter cell based assay.

The VEGF mRNA precipitation results are rather confusing. It appears that the input amounts of each segment differ substantially: is it by chance that the positive IP is the most abundant? The input of some segments are already lower than the amount of positive segment pulled down, making it hard to judge whether the TRS binding was specific to this segment. Should TRS binding to anticodon motif be important for VEGF translation, the overexpression of tRNA(Thr)-anticodon stem or threonine depletion should counteract this activity. Has this been tested?

Reviewer #4 (Remarks to the Author):

The authors have conducted two morpholino-based knockdowns and analysed the resultant vascular phenotypes. Overall, the methods, controls and rescues are all appropriate however I have a few suggestions for improvement:

- 1) How did the authors determine which zebrafish gene was the ortholog of TRS and 4EHP? While an obvious eif4e2 gene is annotated in the zebrafish genome (<http://zfin.org/ZDB-GENE-050327-59>), currently, no zebrafish gene is annotated with "trs". The authors should describe how they identified the zebrafish ortholog, include either the ZFIN or ENSEMBLE number for the gene investigated and use the appropriate zebrafish nomenclature if available.
- 2) The numbers for each morpholino experiment are very low. I would expect the number of animals analysed to be in the 20-30 range not the 5-10 range in this study. This is especially concerning in the TRS morpholino study (Figure 5E,F) which shows only a very minor phenotype. Presumably, with numbers this low these experiments have only been done once and should be repeated to determine consistency.
- 3) The statistical methods for determining significance in Figure 5F, H and Supplementary Figure 6D are not stated.
- 4) While the 4EHP MO appears to be working well, the same cannot be said for the TRS MO. As the TRS morphant phenotype is, at best, subtle, I would suggest removing the TRS data from figure 5 (Fig 5E-F) and presenting the 4 EHP MO ISV phenotype (Sup Fig 6d) in the main figure as this is more convincing data.
- 5) The authors should consider trying another TRS splice-blocking MO that may be more effective.
- 6) Figure 5F, H and Supplementary Figure 6D - standard deviation would be a more appropriate error bar to present than SEM.
- 7) In the methods, the authors should include the dose of MO injected rather than the concentration of the injection solution.

8) While not essential, whole mount in situ hybridisation using probes against the TRS and 4EHP genes would determine whether the spatiotemporal expression supports the phenotypes observed. Looking at published eif4e2 data (<http://zfin.org/action/figure/all-figure-view/ZDB-PUB-040907-1?probeZdbID=ZDB-EST-041111-328>) it appears to be strongly expressed in the somatic muscle and head during ISV and Ct vessel formation, supporting the authors hypothesis.

Description of changes in the revised manuscript:

Reviewer #1 (Remarks to the Author):

This manuscript represents a paradigm shift in our understanding of eukaryotic translation and when acceptable is clearly suited for Nature. As may be anticipated with a new idea, there are a variety of concerns given the different view point. Studies on the mechanism of eukaryotic initiation began with individual components and then the establishment of a pathway followed by in vivo and genetic experiments. These studies seem to have started with the big picture and worked towards some of the smaller details.

At the very outset, if one is to propose that TRS, eIF4E2 and eIF4A are to form a functional complex that replaces eIF4F, where is the evidence (i.e. mix the proteins together and show they all elute as a high molecular weight peak; MW about 200,000)? Secondly, the enzymatic properties of eIF4F are quite different from those for eIF4A; Is this also true for the authors' new complex?

Response: We greatly appreciate the comments of Reviewer #1. We should mention that we came to realize what TRS may play a role in regulating translation initiation following structure determination of the TRS UNE-T and 4EHP complex. Since then, we have gradually elucidated the role of TRS in regulating translation initiation by interacting with other factors such as eIF4A, mainly through cell biological and *in vivo* experiments. We could have now acquired sufficient data to confirm its novel role and function, as described in the manuscript.

Based on the reviewer's comments regarding the formation of the eIF4F-like complex at the protein level, we first purified full-length TRS with a Strep tag at the C-terminal end following expression in HEK293T cells. We would like to note that full-length TRS was

difficult to purify to homogeneity due to its flexible N-terminal region, including UNE-T. 4EHP was purified without a tag as described in the manuscript. We then evaluated the interaction between purified TRS, which forms a dimer via the catalytic domain¹, and 4EHP using *in vitro* Strep-Tactin pull-down assay. The results showed that unlike the isolated UNE-T, full-length TRS interacts weakly with 4EHP (see the figure below, and Supplementary Fig. 1d in the revised manuscript), suggesting that the N-terminal UNE-T may not be fully exposed in the dimeric form of TRS *in vitro*. It is possible that formation of the cellular TRS-4EHP complex might involve a conformational change of the full-length TRS to fully expose UNE-T. We have described these results in the revised manuscript.

In vitro reconstitution of a stable complex of full-length TRS, 4EHP, and eIF4A was difficult due to the intrinsic biophysical properties of TRS as mentioned above. To enhance the stability of the complex, we performed cross-linking. Reaction mixtures containing 20 μ g of purified TRS-Strep, eIF4A, and 4EHP in a total volume of 100 μ L were incubated with 0.0025% (v/v) glutaraldehyde in conjugation buffer (100 mM HEPES, pH 7.5, and 150 mM NaCl) for 5 min at 37°C. The reaction was terminated by addition of 10 μ L of 1 M TRIS-HCl, pH 8.0. The cross-linked proteins were loaded onto Strep-Tactin resin for 30 min at 4°C,

washed thoroughly five times with ice-cold wash buffer (50 mM TRIS-HCl, pH 7.5, 300 mM NaCl, 1 mM EDTA, and 2% NP-40), and eluted with Strep elution buffer (100 mM HEPES, pH 8.0, 150 mM NaCl, and 1 mM EDTA supplemented with 2.5 mM desthiobiotin). The eluted sample was analyzed using denaturing SDS-PAGE and immunoblotting. As shown in the figure below, a cross-linked band corresponding to the covalent TRS-4EHP-eIF4A complex was clearly visible in the monomeric state (stoichiometry 1:1:1) and dimeric state (stoichiometry 2:2:2) following denaturing SDS-PAGE, indicating that they do indeed form a complex. Although we demonstrated the direct interaction between full-length TRS and 4EHP, and trimeric complex formation including eIF4A *in vitro*, we suspect that other factors might be additionally required for the formation of a stable TRS-4EHP-eIF4A complex *in vivo*. Since the main focus of this work was to report the novel functional connection between a protein synthesis enzyme (TRS) and a specific form of an initiation factor (4EHP), we believe that the exact number and identity of additional factors should be systematically investigated in-depth in future studies. For this reason, we have not included the cross-linking results in the revised manuscript, and instead include them in this response letter.

Regarding the enzymatic activity of eIF4A, we evaluated the effect of TRS on the ATPase activity of eIF4A. The results showed that the ATPase activity of eIF4A was enhanced by TRS, but not by LRS (see the results below). Thus, these results further suggest that TRS may play a similar role to eIF4G in the eIF4F complex, as deduced in a previously reported study showing that the ATPase activity of eIF4A is augmented by eIF4G².

Figure. Enhancement of the ATPase activity of eIF4A by TRS. The effect of TRS on the ATPase activity of eIF4A was determined using a malachite green assay (R&D Systems). The specific activity was determined using a phosphate standard curve, and all experiments were performed in triplicate with similar results.

Much of the authors' data reflect various pulldown experiments followed by Western blotting. While specificity may be seen in that eIF4A2 is pulled down (Figure 1A) and eIF4A1 is not, it is not clear how many other interacting proteins might have been missed because they were not blotted for. That is, if the pulldown was subjected to SDS gel electrophoresis, how many

protein bands would be visualized? Secondly, it is not clear from either the figure legends or methods as to how often the experiment was performed nor are error bars evident in many of the figures (i.e. what is the variation between blots and how are they normalized?).

Response: We assume that eIF4A2 mentioned here by the Reviewer is eIF4E2, which is 4EHP. As described in the original manuscript, we performed affinity purification of Strep-tagged TRS-interacting proteins (see the figure below) and analyzed them by LC-MS. From the analysis, we identified 434 proteins as potential TRS-associating proteins, directly or indirectly, including proteins involved in post-transcriptional regulation, mRNA metabolic processes, and translation (Supplementary Fig. 1a, b). We also carried out yeast LexA-B42 two-hybrid screening and identified four proteins as potential TRS-interacting proteins, including TRS itself (Supplementary Fig. 1c). Based on these results, we decided to focus to the interaction of TRS with 4EHP because both analyses predicted the potential interaction of the two proteins, and because both are commonly involved in translation processes^{3, 4, 5, 6}.

We further examined the potential interaction of many other ARSs with 4EHP and found that only TRS engaged in specific binding to 4EHP (Fig. 1b). The specificity of the cellular interaction between TRS and 4EHP was further demonstrated by BiFC experiments (Fig. 1e). These new data have been added to the revised manuscript.

Regarding the concerns about reliability of the data raised by the Reviewer, all experiments were repeated independently at least three times, and data in the revised manuscript are representative of experiments with similar results, shown as means \pm SEM.

Specific concerns

Q1. *The TRS is a dimer. How representative are the results achieved with a monomer (mostly for pieces of TRS such as VN, VC, UNE, etc. as either pull downs or crystal structure)?*

Response: We conducted most of the pull-down and immunoprecipitation assays using full-length TRS, and used the separated domains to determine the part responsible for the interaction with 4EHP.

For bimolecular fluorescence complementation (BiFC) assays, a useful method for monitoring cellular protein-protein interactions, we also used full-length proteins. The Venus N-terminal domain (VN) and Venus C-terminal domain (VC) mentioned by the Reviewer are nonfluorescent N- and C-terminal domains of the Venus fluorescent protein. When VN and VC are reconstituted to form the functional Venus protein following interaction of the two proteins of interest, green fluorescence is recovered. The TRS and 4EHP (eIF4E2) pair was the only combination that facilitated formation of the bimolecular fluorescent complex. We further confirmed the specificity of the TRS-4EHP interaction by extending the repertoire of factors tested (Fig. 1e).

To determine the crystal structure of the TRS and 4EHP complex, we had to use UNE-T, the region responsible for the interaction with 4EHP, because UNE-T is flexibly linked to the catalytic domain of TRS. Nonetheless, the structural information obtained for the complex provided insight into the novel function of TRS in the regulation of translation initiation, and this function was further validated by comprehensive *in vitro* experiments and *in vivo* phenotype analysis.

Q2. *There does not seem to be a mathematical representation of the strength/relative affinity of TRS for other proteins. It would be of value to have some better idea as to what this might be as this is not all that evident from supplementary figure 1.*

Response: We acknowledge the Reviewer's comments. As emphasized in the main text, this work revealed a novel translational initiation complex involving TRS (an aminoacyl-tRNA synthetase) and 4EHP (an isoform of eIF4E). We do not yet know how many other unknown factors may be required for a fully functional complex. In this particular work, we focused on determining the structure of the binary complex of TRS and 4EHP, and its functional implications in translational initiation. We believe other factors required for full activity and their interactions with the rest of the components are subjects for future research.

Nonetheless, based on the reviewer's suggestion, we examined the direct interactions of TRS with eIF4A, and also with PABP, using *in vitro* pull-down assays. Purified eIF4A and PABP proteins (without a tag) were sticky and bound non-specifically to Ni-NTA agarose resin and glutathione Sepharose resin. To avoid these problems, a reaction mixture containing purified full-length TRS-His and eIF4A was incubated with eIF4A antibody, and pulled down with Protein A/G PLUS-agarose beads, and co-purification of the two proteins as a complex was determined by immunoblotting analysis. The results confirmed the direct interaction of the two factors (Supplementary Fig. 10a).

To test the interaction between TRS and PABP, we purified full-length TRS-Strep and His-GST-PABP, loaded the reaction mixture onto Strep-Tactin resin, eluted with Strep elution buffer, and co-purification of the two factors as a complex was determined by immunoblotting analysis. The results also confirmed the direct interaction of TRS and PABP (Supplementary Fig. 12b). These novel insights have been added and described in the revised manuscript.

Q3. *The authors often describe pull down results as indicating “a specific interaction” if one or two other proteins are not seen. This is insufficient (see above).*

Response: We understand that “a specific interaction” can be better supported by using a greater number of control proteins for comparison. Thus, we further validated the specificity of the TRS-4EHP interaction using various aminoacyl-tRNA synthetases (ARSs) in the revised manuscript. Among the 13 different ARSs tested, only TRS engaged in an interaction with 4EHP (Fig. 1b). In the case of eIF4E homologues in humans, only eIF4E2 (4EHP) interacted with TRS (Fig. 1a). We also confirmed the specific interaction between TRS and 4EHP using bimolecular fluorescence complementation (BiFC) assays (Fig. 1e) in the revised manuscript.

Q4. *The TRS-4EHP interaction is cited as specific for vertebrates, yet there is only a single non-vertebrate tested (fly). It would be useful to extend this to at least two other negative species.*

Response: Following the Reviewer’s comment, we further evaluated the interaction between TRS and 4EHP from yeast and nematode, and observed no interaction in these two lower eukaryotic organisms. These new results have been added in the revised manuscript (Fig. 3a, b, f, g).

Q5. *Using AlaRS as a control, the RNA pulldown with TRS was reduced from 7,255 (IgG control) to 2,928. How much more of a reduction might be seen if additionally ProRS and IleRS RNA pulldowns were done (i.e. other tRNA synthetases that have either A or C as the first or second nucleotide in their codon)?*

Response: Following the Reviewer’s comments, we repeated the mRNA enrichment

experiments not only with anti-TRS antibody, but also with anti-PRS and IRS antibodies, and compared the enriched mRNAs. Similar to the results obtained using AlaRS as a control, the number of TRS-enriched mRNAs was reduced from 7,708 (IgG control) to 2,694 and 2,866 after subtraction of PRS- and IRS-enriched mRNAs, respectively. Furthermore, functional annotation clustering analysis revealed that a large proportion of the enriched GO terms were related to vertebrate system development. These results have been described in the revised manuscript (Supplementary Fig. 3).

Q6. *The use of either “TRS expressing cells” or siTRS transfected cells seems problematic. First, there is no quantitation as to the level of overexpression of TRS and secondly, the affect observed is barely two-fold over empty vector control. With respect to the use of siRNAs, it is anticipated that protein synthesis requires all 20 synthetases. As the measurement of luciferase is 36 to 72 hours after siRNA treatment, it is not clear what level of protein synthesis is actually occurring. Is there the expected decrease in protein synthesis (as might be measured by S35 methionine incorporation)?*

Response: We understand the Reviewer’s concerns. Accordingly, we have added the quantitation of TRS overexpression to the revised manuscript. Regardless of the degree of overexpression, we believe that the mild positive effect following TRS overexpression could be due to the presence of endogenous TRS that is sufficient for translation. Regarding the effect of anti-TRS siRNA on protein synthesis, there are previous reports showing that ARS knock-down has minimal effect on global translation under normal conditions over a long duration^{7, 8}. Consistent with previous reports, we also observed little effect of TRS knock-down on global translation in our experimental conditions, as determined by S³⁵ Met-incorporation assays (newly added to the revised manuscript; Supplementary Fig. 11c). These results are also consistent with those of the puromycin incorporation assay described

in the original manuscript.

Q7. *Given the apparent effectiveness of the TRS knockdown (supplementary figure 4A), it is surprising that the level of VEGF expression is only reduced 30 to 40% in HEK293T cells (figure 5A).*

Response: Without doubt, protein expression levels are determined by the combined results of gene transcription, translation, and protein turnover, and these activities can differ between cell types. While VEGF levels showed high dependency on TRS-mediated translation control in macrophages (DTHPs in our study), vascular smooth muscle cells (VSMCs in our study) and lung fibroblasts (WI-26 in our study; Fig. 5a), TRS dependency was less significant in HEK293T cells, as pointed out by the Reviewer. Given that the TRS dependency of cellular VEGF levels varies depending on cell type, it is not unexpected that control of VEGF expression levels by TRS may be less significant in HEK293T cells.

Q8. *In supplementary figure 7 it is shown that m⁷GTP-Sepharose can pulldown TRS (via its interaction with 4EHP). What other proteins are also pulled down with 4EHP under these conditions?*

Response: When we performed experiments related to the results shown in Supplementary figure 7, we had no idea whether TRS functions similarly to eIF4G or eIF4F, and therefore did not know exactly what to check for in the pull-down samples. eIF4A was later examined as a component of the TRS-mediated complex, and in subsequent m⁷GTP-Sepharose pull-down experiments, we showed that TRS and 4EHP form a distinct complex separate from the eIF4G-eIF4E complex, and both complexes contain eIF4A (Fig. 6d, e). Although we acknowledge the Reviewer's comments, given the order of our experiments and discoveries

in our study, we believe it is better to keep the current dataset. However, to strengthen the original results, we have also checked AlaRS and KRS (as negative controls) and eIF4E (as a positive control) in the cap-dependent complex consisting of 4EHP and TRS, and added the new data to the revised manuscript (Supplementary Fig. 9).

Q9. *Figure 7F is supposed to show an interaction with eIF3 and TRS. However, only a few eIF3 subunits appear visible. Why are not all of the eIF3 subunits pull down?*

Response: Consistent with the Reviewer's comments, we observed that all eIF3 subunits except for eIF3A and I were immunoprecipitated with TRS in pull-down experiments following co-expression of TRS-Strep with each of the FLAG-tagged eIF3 subunits (see the results below). However, we believe that some of these eIF3 subunits could be pulled down with TRS indirectly via direct TRS-binding subunits. To identify potential direct TRS binders, we conducted *in vitro* pull-down assays with GST-TRS and each of the eIF3 subunits expressed as FLAG-tagged proteins using glutathione-Sepharose. To exclude indirect binders, we thoroughly washed the TRS-binding mixture bound to the beads, and found that only the subunits eIF3B, D, F, and L survived the stringent washing step and co-precipitated with GST-TRS (Fig. 7e). Endogenous TRS was immunoprecipitated from WI-26 cells with a mouse anti-TRS antibody and examined for associated eIF3 subunits using antibodies recognizing each of the subunits (Fig. 7f). We also performed intensive washing steps for these experiments, and again found that subunits B, D, F, L (and E) co-purified with TRS, further confirming the above results.

Figure. Co-immunoprecipitation assay of TRS-Strep co-expressed with each of the FLAG-tagged eIF3 subunits in 293T cells. TRS-Strep was purified using Strep-Tactin resin, and co-precipitated eIF3 subunits were determined by immunoblotting with anti-FLAG antibody.

In addition, we also analyzed the interaction between eIF3 and TRS at the protein level. We first transfected each of the eIF3 subunits (B, D, F, and L) tagged with Strep, which demonstrated interaction with TRS in *in vitro* pull-down assays with GST-TRS (Fig. 7e) and endogenous interaction assays (Fig. 7f) in HEK293T cells, and subsequently attempted to purify each eIF3 subunit using Strep-Tactin resin. We found that most of the eIF3 subunits except for A, G, I, and J were associated with the transfected subunit D (see the results below, Figure a, b). Thus, we used the eIF3 subunits associated with subunit D to assess the interaction with TRS *in vitro*. Purified GST-TRS was incubated with the eIF3 subunits, pulled down with glutathione-Sepharose beads, and washed intensively. Co-precipitation of the eIF3 subunits with GST-TRS was then determined by immunoblotting with antibodies recognizing each of the subunits. The results showed that TRS interacts with the same subunits (B, D, E, F, and L; see the results below, Figure b) as those implied by the results of *in vitro* pull-down assays (Fig. 7e) and endogenous interaction assays (Fig. 7f).

Reviewer #2 (Remarks to the Author):

In this manuscript Jeong et al. have shown that human TRS (hTRS) has acquired additional functionality of regulating translation of other genes involved in a variety of physiological processes, the majority being related to vertebrate system development. Using pull-down and co-IP assays extensively, they have identified different domains/regions of hTRS that interact with specific set of mRNAs (e.g., VEGF) and other protein factors (4EHP) to form an initiation complex that is analogous to the canonical eIF4F complex, but functions independently of the latter. The authors have also solved the co-crystal structure of (4EHP + hTRS UNE-T) that has identified the specific interactions between these two proteins, and

corroborates other findings. They have further demonstrated that such a regulation of translation by TRS is specific to vertebrate lineage, since fruit fly TRS fails to form such a complex. Their functional assays show the effect of knocking down of TRS–4EHP initiation machinery on target gene expression and its phenotypic manifestations. The authors have performed significant amount of work by approaching the problem from structural, mechanistic and cell biological perspectives. Overall, the current study has identified a hitherto unknown function of hTRS in particular, and vertebrate TRS in general, in the translational regulation of a multitude of metabolic processes, suggesting that such a gain-of-function in vertebrate lineage have played a role in the delineation of vertebrates from invertebrates. The results are novel and provide a totally new dimension to the problem of translation regulation in vertebrates.

Concerns

Q1. Page 10, Supplementary fig 3a: The changes observed in luciferase assay upon changing the UGU (Thr) anticodon-like loop to even AAA (Phe) anticodon-like loop are less than 5%, when compared to luciferase activity in the case of UGU (Thr) anticodon-like loop. This implies that recognition of anticodon-like loop structure by TRS may not be strictly sequence specific. The authors should check the effect of loop proximity to the translation start site (i.e. the position of the loop in 5' UTR) as well as loop length on translation regulation (luciferase expression). Furthermore, they should also check for the presence of such loops in other genes, at least in the case of ANG gene that was tested in the study.

Response: Following the reviewer's suggestion, we changed the loop position both upstream and downstream of the translation start site and examined the effect on reporter

expression. As the reviewer predicted, the loop position from the translation start site was critical for recognition by TRS (Supplementary Fig. 4b). We have further evaluated the effects of loop length on translation regulation in the revised manuscript, and observed that the length of the loop is also important for recognition by TRS (Supplementary Fig. 4c). We also identified the potential TRS-recognizing loop located in the 5' UTR of *ANG* mRNA, and similar results were obtained with a loop incorporated upstream of the luciferase gene, as detailed in the revised manuscript (Supplementary Fig. 4d, e).

Q2. *In vitro pull-down assay needs to be checked to confirm direct interaction between TRS and PABP because the latter's detection in co-IP with TRS ABD can also be due to interaction of mRNAs with ABD.*

Response: Following the reviewer's suggestion, we first evaluated whether the TRS interaction with PABP is mediated by mRNA using RNase treatment, as outlined in the revised manuscript. PABP-FLAG was co-expressed with TRS-Strep in 293T cells and immunoprecipitated with anti-FLAG antibody, and co-precipitated TRS was detected by immunoblotting with anti-Strep antibody. No change in the interaction between the two proteins was observed following treatment with RNase, suggesting that TRS may directly interact with PABP in cells (Supplementary Fig. 12a). To confirm the direct interaction of TRS and PABP, we purified TRS-Strep and His-GST-PABP, loaded the mixture onto Strep-Tactin resin, washed thoroughly, and eluted with Strep elution buffer for immunoblotting analysis. The results revealed that the two proteins directly interact with each other (Supplementary Fig. 12b). These results have been described in the revised manuscript.

Q3. *TRS LLL mutant also can be probed for interaction with eIF4A, PABP and eIF3 to rule*

out any overlap between binding sites of these factors and that of UGU anticodon-like loop.

Response: We evaluated whether the TRS ABD site binding to the anticodon^{Thr}-like loop overlaps regions interacting with eIF4A, PABP, and eIF3. The results clearly showed that mutation of residues in the TRS ABD that is critical for binding to the anticodon^{Thr}-like loop did not disturb the interactions with other translation initiation factors. These results have been added to Supplementary Fig. 10b in the revised manuscript.

Q4. *In the model, emphasis should be put on the importance of ABD's recognition of the UGU anticodon-like loop, leading to the binding of other factors to form the initiation complex.*

Response: We have described this point in the revised manuscript.

Q5. *The authors need to tone down their claim about invertebrate-to-vertebrate transition. I would suggest that this point can be emphasized in 'Discussion' and not in 'Abstract'.*

Response: As suggested, we have amended both the Title and Abstract in the revised manuscript.

Minor comments

Q1. *Page 3, para 2: "Cell condition-specific...eIF4 isomers mediate...sequesters eIF4E." Change "isomers" to "isoforms".*

Response: We have corrected this in the revised manuscript.

Q2. Page 4, para 2: *“The machinery includes...to eF4F composed...and eIF4A.”* Change “eF4F” to “eIF4F”.

Response: We have corrected this in the revised manuscript.

Q3. Page 6, para 3: *What is meant by “dorsal and lateral surfaces of eIF4E”? It is better to explain these terms (preferably by a supplementary fig, if possible) for the convenience of the general reader.*

Response: As suggested, we have explained these terms in Fig. 2b of the revised manuscript.

Q4. Page 6, para 3: *“Furthermore,...interacting with eIF4E...”* change “interacting” to “interact”.

Response: We have corrected this in the revised manuscript.

Q5. Pages 6/7, paras 3/1: *“However, TRS does not appear...immediately after Asp78 (Fig. 1f,g).”* This point is not clear! It is better to elaborate on this point a bit. How does TGS domain beginning immediately after Asp78 not lead to TRS not utilizing the non-canonical motifs and auxiliary sequences? The reference of Fig 1f,g does not seem to support/clarify this point at all. Moreover, it is not clear whether TRS possesses non-canonical motifs and auxiliary sequences.

Response: We have clarified the fact that TRS does not possess the non-canonical and auxiliary motifs in the revised manuscript.

Q6. Page 8, para 2: *“Mutation of ...motif of zebrafish 4EHP...”* change “zebrafish 4EHP” to “zebrafish TRS”.

Response: We have corrected this in the revised manuscript.

Q7. Page 9, para 2: *“Remarkably, most genes...development of nervous, skeletal, and circulation systems...”* Nervous and circulatory systems are not unique to vertebrates! Thus, the sentence needs to be modified.

Response: We have modified this sentence in the revised manuscript.

Q8. Page 11, para 2: *“A slight reduction...was silenced (Supplementary Fig. 4a).”* Explanation? Error bars?

Response: It has been reported that silencing of eIF4G reduces VEGF production more than does silencing of eIF4E⁹. We have added error bars in the revised manuscript.

Q9. Page 11, para 2: *“Silencing of TRS...by oxygen tension (Supplementary Fig. 4e).”* What is the relevance of this aspect in the context of the study?

Response: VEGF is generally induced during hypoxia. In addition, as described in the Introduction section, together with oxygen-regulated hypoxia-inducible factor 2 α and RNA-binding protein RBM4, 4EHP regulates global hypoxic protein synthesis during hypoxia⁴. Therefore, whether TRS and 4EHP-mediated translation initiation is affected by oxygen availability should be checked in this study.

Q10. *Page 12, para 1: “Tube formation and...TRS (M60K)-expressing cells (Fig. 5d and Supplementary Fig. 5d).” Enhancement in the case of M60K is seen for both tube length and cell migration, but to a lesser extent than wild-type. Hence, the statement is incorrect and needs to be modified accordingly.*

Response: We thank the Reviewer for pointing out this incorrect statement. We have modified this statement “Tube formation and...TRS (M60K)-expressing cells (Fig. 5d and Supplementary Fig. 5d).” to “Tube formation and cell migration of HUVECs were significantly enhanced by supernatants from TRS-transfected WI-26 cells, but to a lesser extent by supernatants from 4EHP-binding-defective TRS (M60K)-expressing cells (Fig. 5d and Supplementary Fig. 6f, g).” in the revised manuscript.

Q11. *Page 16, para 2: “During evolution,...complexity of higher organisms.” Whether increasing complexity resulted in the acquisition of additional functionalities or the latter resulted in increased complexity? This is a contentious and debatable statement, and therefore needs to be modified accordingly.*

Response: We have modified the statement from “During evolution, ARSs appear to have expanded their functions beyond their catalytic role in protein synthesis through the acquisition of new domains with unique structural features to cope with the increasing biological complexity of higher organisms.” in the original manuscript to “Higher eukaryotic ARSs have undergone stepwise molecular evolution, including the addition of new domains with unique structural features that correlate with the increasing biological complexity of higher organisms.” in the revised manuscript.

Q12. *Page 17, para 4, Supplementary fig 9: Since this is a dataset generated in the present work, it should be reported in the 'Results' section, and not merely in the 'Discussion' section.*

Response: We appreciate the Reviewer's thoughtful comments. As described in the original manuscript, our observation of the emergence of murine 4EHP and VEGF at embryonic day 10.5 (Supplementary Fig. 13 in the revised manuscript) coincides with previously reported results showing that early endothelial cell differentiation followed by migration and tube formation occurs at embryonic day 10.5 during murine embryonic development, along with pulmonary vessel formation. These findings led us to suggest the possibility that expression of 4EHP might be a cue for TRS to assemble a specific translation initiation complex. Nonetheless, we believe that the detailed mechanism underlying the temporal and spatial collinearity of TRS and 4EHP in translation initiation requires further investigation, hence we would prefer to leave this statement in the Discussion section if at all possible.

Q13. *Crystallographic table: round off the unit cell dimensions to 2 decimal places; mention the number of unique reflections in the highest resolution shell; mention the resolution range for the highest resolution shell used in refinement; mention the number of reflections used in refinement; mention avg B-factor for protein, ligand and water separately; change "Geometry (%)" to "Ramachandran statistics (%)"; mention what % of reflections were used for Rfree calculation during refinement.*

Response: In line with the reviewer's suggestion, we have altered the crystallographic data statistics in Supplementary Table 1 in the revised manuscript.

Q14. *Fig 1h: Provide the standard deviation value for Kd.*

Response: ITC is a technique that directly measures the heat exchange accompanying a

chemical or biochemical reaction (or physical process). ITC investigates the energetics of ligand binding to biological macromolecules and provides a complete thermodynamic characterization of the macromolecule-ligand interaction, thereby generating a reliable measure of the binding affinity, as well as changes in enthalpy and entropy during the process. Each ITC analysis utilizes a precisely controlled series of steps to calculate thermodynamic parameters, including the subtraction of experimental controls. For these reasons, ITC results do not show error bars on the figure, as can be seen in numerous published papers. However, ITC analysis provides an error range for each parameter (except for the dissociation constant, K_d) for each experiment. To address concerns raised by the Reviewer, we have included an association constant (K_a) and its associated error range in the revised manuscript (Fig. 1h). In addition, we performed the ITC experiments three times, and the data presented are representative of experiments with similar results. These descriptions have been added in the figure legend of the revised manuscript.

Q15. *Fig 2a: The Fo-Fc map should be an unbiased map generated before modeling of the ligand.*

Response: Fig. 2a showing the Fo-Fc map was generated using the procedure recommended by the reviewer.

Q16. *Supplementary fig 3b: Show quantification of hTRS to show dose-dependent increase in protein level.*

Response: Following the Reviewer's comment, we have shown the dose-dependent increase in TRS expression in cells at the protein level in the revised manuscript (Supplementary Fig. 4g).

Q17. *Supplementary fig 4c: Why do eIF4E bands appear so faint? Error bars?*

Response: We have repeated these experiments and updated the results in the revised manuscript (Supplementary Fig. 5c).

Q18. *Error bars missing in Supplementary fig 4a,c,e.*

Response: We have added error bars in the revised manuscript (Supplementary Fig. 5).

Reviewer #3 (Remarks to the Author):

The authors present a study that proposes a new function of an aminoacyl-tRNA synthetase in stimulating cap-dependent translation of VEGF. Overall, this is an interesting study and has the potential to transform our understanding of how selective translation of VEGF can be regulated by 4EHP. A strength of the study is the structural model that reveals how 4EHP binds to TRS. The validation of this interaction using structure guided mutants is impressive (although mutations in zebrafish TRS did not clearly result in loss of binding (Fig. 3d)). Several weaknesses of the study lowers enthusiasm regarding the validity of the proposed model. Basically, more rigorous experiments are needed to strengthen the findings.

There is a considerable emphasis on interaction models being generated from immunoprecipitation data from the over-expression of epitope tagged proteins in cells. The possibility of indirect binding partners being involved for interactions is not rigorously explored. This is particularly evident in the case of eIF4A and eIF3 binding to TRS. The ability of TRS to bind eIF4A and eIF3 needs to be more rigorously tested using biochemical

and biophysical approaches to substantiate the proposed model whereby TRS functions in place of eIF4G. The VEGF translation reporter assay requires the over-expression of 4EHP and TRS. This raises some concern about how this interaction promotes translation in an endogenous system when components are not over-expressed.

Response: We repeated the zebrafish TRS mutation validation experiments and obtained consistent but more convincing results. The new results have been added to the revised manuscript (Fig. 3d).

We agree with the point raised by the Reviewer that the possibility of indirect binding partners being involved in interactions was not rigorously explored in the original manuscript. We should emphasize that we do not yet know how many other unknown factors may be required for a fully functional TRS-mediated translation initiation machinery. In this particular work, we focused on determining the structure of the binary complex of TRS and 4EHP, and its functional implications in translational initiation. Therefore, we believe that determining how many other factors may be required, directly or indirectly, for full activity, and their interactions with the rest of the components, are subjects for future studies.

Regarding TRS binding to eIF4A, we further evaluated the interaction at the protein level in the revised manuscript. Please see the response to Q3 below for detailed information.

We also analyzed the interaction between eIF3 and TRS at the protein level. We first transfected each of the eIF3 subunits tagged with FLAG, and conducted *in vitro* pull-down assays with GST-TRS. The results confirmed that TRS makes direct interactions with the eIF3 subunits B, D, F and L (Fig. 7e). We further conducted co-immunoprecipitation experiments between endogenous TRS and the eIF3 subunits in HEK293T cells and observed that TRS can also interact with the eIF3 subunit B, D, F and L (also with E) (Fig. 7f). We subsequently attempted to purify each of the eIF3 subunits using Strep-Tactin resin and found that most of the eIF3 subunits except for A, G, I, and J were tightly associated with the

subunit D (see the results below, Figure a, b). Thus, we used the eIF3 subunits enriched with subunit D to assess the interaction with TRS *in vitro*. The purified GST-TRS was incubated with the eIF3 subcomplex containing the subunits (B, C, D, E, F, H, K, L, M), pulled down with glutathione-Sepharose beads, washed intensively, and the eIF3 subunits co-precipitated with GST-TRS were then determined by immunoblotting with the antibodies recognizing each of the subunits. The results again showed the direct interaction of TRS with the subunits (B, D, E, F, and L; see the results below, Figure b) as the results above (Fig. 7e, f) although some subunits are missing in this experiment. Combined together, it can be concluded that TRS can make direct interactions with the eIF3 subunits, B, D, F, L (and possibly E).

Regarding the point raised by the Reviewer that the VEGF translation reporter assay requires overexpression of 4EHP and TRS, and that this raises some concern about how

this interaction promotes translation in an endogenous system when components are not overexpressed, this is a reasonable concern and generally applicable to any biological experiments in which the overexpression effects of the factor of interest are monitored. Since the majority of TRS molecules would be occupied during catalysis, only a small portion would be used for translation initiation complex formation. In this work, we also showed that 4EHP is strongly induced at embryonic day 10.5, which would affect its availability to form a translation initiation complex (Supplementary Fig. 13). Based on these facts, we predict that rather than other translation factors, TRS and 4EHP could be the limiting components involved in the formation of the translation initiation complex.

Detailed comments:

Q1. *As the authors point out, the translation of VEGF can be regulated by hypoxia. It is shown that knockdown of 4EHP still inhibits VEGF translation in reduced oxygen conditions (FIG S4). It should be noted that the authors need to add error bars to all the data presented in Fig. S4 to show that the data are statistically relevant.*

Response: We thank the reviewer for this comment. We have added error bars in the revised manuscript.

Q2. *The translation of VEGF has been shown to be sensitive to both eIF4E and eIF4G levels, in both normal and hypoxic conditions (Int J Cancer J Int Du Cancer 1996; 65:785-90; PMID:8631593; Mol Cell. 2007;28:501–512). The authors need to confirm that eIF4E and eIF4G knockdown do in fact reduce VEGF translation in hypoxic conditions in their system.*

Response: In line with the reviewer's comment, we have confirmed that knock-down of eIF4E or eIF4G decreased the translation of VEGF under hypoxic conditions, and the results are included in the revised manuscript (Supplementary Fig. 5g).

Q3. *The claim that eIF4A binds directly to TRS is most unexpected. This finding should be more rigorously investigated to substantiate the proposed model. Is the translation of VEGF sensitive to eIF4A inhibition by PDCD4 and hippuristanol? The in vitro binding assays (ITC and pull-downs using purified recombinant full-length factors) should be repeated with the addition of eIF4A. A mutant TRS that cannot bind to eIF4A should also be generated and tested in the TRS over-expression VEGF reporter cell-based assay.*

Response: Regarding the question of whether the translation of VEGF is sensitive to eIF4A inhibition by PDCD4 or hippuristanol, the results showed that both hippuristanol (Figure a) and PDCD4 (Figure b) inhibit VEGF translation, as illustrated in the figure below.

Similar to the reviewer, we were also surprised to observe an interaction between TRS and eIF4A. Although we tried many different approaches to obtain a clear picture of the interaction, most experiments did not produce desirable results, mainly due to the challenging biochemical properties of full-length TRS (perhaps due to the flexible linker between the catalytic domain and UNE-T) and eIF4A. After numerous attempts, we finally purified full-length TRS-His and eIF4A (without a tag), and the mixture was reacted with anti-

eIF4A antibody, mixed with Protein A/G PLUS-agarose beads, washed extensively, and subjected to immunoblotting analysis. The results confirmed the direct interaction of TRS and eIF4A (Supplementary Fig. 10a). These results have been described in the revised manuscript.

We also evaluated the effect of TRS on the ATPase activity of eIF4A. The results showed that the ATPase activity of eIF4A is enhanced by TRS, but not by LRS (see the results below). Thus, these results further indicate that TRS may play a similar role to eIF4G in the eIF4F complex, as previously reported for the ATPase activity of eIF4A, which is augmented by eIF4G².

Figure. Enhancement of the ATPase activity of eIF4A by TRS. The effect of TRS on the ATPase activity of eIF4A was determined using a malachite green assay (R&D Systems). The specific activity was determined using a phosphate standard curve generated. All experiments were performed in triplicate with similar results.

We attempted to identify a TRS mutant that does not interact with eIF4A. We should emphasize that finding TRS residues that are critical for interacting with eIF4A is very difficult without structural information on the interaction between the two proteins. Nonetheless, using previously reported structural information for each of the proteins, we screened mutants and found that mutations at positions R699, E703, E706, and R707 (mutant #4) in TRS weakened the interaction with eIF4A (see the results below, Figure a). Subsequently, we evaluated the translation efficiency of the luciferase reporter gene in TRS WT- or mutant #4-expressing 293T cells. Luciferase expression was reduced in the mutant-expressing cells compared with WT-expressing cells (see the results below, Figure b), revealing the importance of the TRS interaction with eIF4A in regulating translation initiation. Although we demonstrated that disruption of the interaction between TRS and eIF4A reduces the translation effectiveness, we do not believe that the mutant identified and employed in this experiment is the best representative for this experimental purpose. This issue should be dealt with in future research, including systematic biochemical analysis of interactions between the two proteins, combined with cell biological functional studies. Thus, we have not included these preliminary results in the revised manuscript.

Figure. Effect of the TRS-eIF4A interaction on translation of a reporter gene. The anticodon Thr-like loop (5' UTR-

167) containing the reporter gene was co-expressed with *Renilla* luciferase to monitor TRS-dependent and nonspecific translation, respectively. The dual luciferase assay was performed on 293T cells expressing WT TRS-Strep and the eIF4A-interacting deficient mutant #4-Strep. *** $p < 0.001$, **** $p < 0.0001$ vs. control (Cont). Values are means \pm SEM of three independent experiments.

Q4. *The VEGF mRNA precipitation results are rather confusing. It appears that the input amounts of each segment differ substantially: is it by chance that the positive IP is the most abundant? The input of some segments are already lower than the amount of positive segment pulled down, making it hard to judge whether the TRS binding was specific to this segment. Should TRS binding to anticodon motif be important for VEGF translation, the overexpression of tRNA(Thr)-anticodon stem or threonine depletion should counteract this activity. Has this been tested?*

Response: To confirm the interaction of TRS with VEGF mRNA, we more carefully loaded the same amount of VEGF mRNA segments and re-analyzed the TRS-bound mRNA as before, and obtained consistent results with those shown in the original manuscript. The results have been updated in the revised manuscript (Fig. 4d).

We also examined the Reviewer's suggestion to check the effects of threonine depletion and tRNA^{Thr}-anticodon stem overexpression on the translation of VEGF mRNA. Since it is already known that amino acid depletion can enhance the expression of VEGF regardless of amino acid type^{10,11}, we did not expect that VEGF translation would be differentially affected by the depletion of different amino acids. When we compared that VEGF expression levels between the leucine- and threonine-depletion conditions, we did not see apparent difference between the two conditions (see the results below, Figure a). We also compared the effects of threonine and alanine tRNA-anticodon stem overexpression on VEGF translation and

observed little effect (see the results below, Figure b). Although the experiments did not provide the results the reviewer suggested, perhaps, the expression level or TRS-binding affinity of the ectopically expressed tRNA-anticodon stem loop have not been sufficiently high enough to take off TRS from the TRS-mediated translation initiation complex in which TRS was multiply associated not only with VEGF mRNA but also with many translation factors (as shown above).

Reviewer #4 (Remarks to the Author):

The authors have conducted two morpholino-based knockdowns and analysed the resultant vascular phenotypes. Overall, the methods, controls and rescues are all appropriate however I have a few suggestions for improvement:

Q1. *How did the authors determine which zebrafish gene was the ortholog of TRS and 4EHP? While an obvious eif4e2 gene is annotated in the zebrafish genome (<http://zfin.org/ZDB-GENE-050327-59>), currently, no zebrafish gene is annotated with “trs”. The authors should describe how they identified the zebrafish ortholog, include either the ZFIN or ENSEMBLE*

number for the gene investigated and use the appropriate zebrafish nomenclature if available.

Response: On the ZFIN website, the threonyl-tRNA synthetase gene is annotated 'trs' (not 'trs'), and its ZFIN ID is ZDB-GENE-041010-218 with the link <http://zfin.org/ZDB-GENE-041010-218>. It is also annotated in Ensembl with gene ID ENSDARG00000013250, with the link http://asia.ensembl.org/Danio_rerio/Gene/Summary?g=ENSDARG00000013250;r=5:44346691-44371177;t=ENSDART00000034523. We have included the gene IDs for trs and eif4e2 in the Methods section in the revised manuscript.

Q2. *The numbers for each morpholino experiment are very low. I would expect the number of animals analysed to be in the 20-30 range not the 5-10 range in this study. This is especially concerning in the TRS morpholino study (Figure 5E,F) which shows only a very minor phenotype. Presumably, with numbers this low these experiments have only been done once and should be repeated to determine consistency.*

Response: We thank the reviewer for the comments. Regarding comments (4) and (5), we found a more effective trs morpholino (trs-i6e7 MO) that worked better than the previous one described in the original manuscript. Using this new trs-i6e7 MO, we have increased the number of animals and repeated each experiment with at least 20 embryos per condition (up to 30 embryos in some cases). The results are included in the revised manuscript (Fig. 5e–h and Supplementary Fig. 7d).

Q3. *The statistical methods for determining significance in Figure 5F, H and Supplementary Figure 6D are not stated.*

Response: We have stated the statistical methods (one-way ANOVA for Fig. 5f and 5h, and student t-test for Supplementary Fig. 7d) in the Methods section of the revised manuscript.

Q4. *While the 4EHP MO appears to be working well, the same cannot be said for the TRS MO. As the TRS morphant phenotype is, at best, subtle, I would suggest removing the TRS data from figure 5 (Fig 5E-F) and presenting the 4 EHP MO ISV phenotype (Sup Fig 6d) in the main figure as this is more convincing data.*

Response: Since we now have more convincing data for the TRS morphant phenotype, we have replaced the original data with the new data in the revised manuscript. We are grateful for the Reviewer's suggestion to improve the data quality.

Q5. *The authors should consider trying another TRS splice-blocking MO that may be more effective.*

Response: Based on the reviewer's comments (Q4) and (Q5), we designed and tested several new candidate trs morpholinos to find a more effective one. We eventually identified a morpholino (trs-i6e7 MO) that worked better than the one described in the original manuscript. Since the knock-down of TRS using the trs-i6e7 MO elicited more overt cerebrovascular phenotypes, we have replaced the original data with the new data obtained using the trs i6e7 MO in the revised manuscript (Fig. 5e, f).

Q6. *Figure 5F, H and Supplementary Figure 6D - standard deviation would be a more appropriate error bar to present than SEM.*

Response: Following the reviewer's suggestion, we have changed the error bars to

standard deviation (SD) for all graphs in the revised manuscript (Fig. 5f, h; Supplementary Fig. 7d).

Q7. *In the methods, the authors should include the dose of MO injected rather than the concentration of the injection solution.*

Response: As suggested, we have described the dose of MO injection in the Methods section of the revised manuscript.

Q8. *While not essential, whole mount *in situ* hybridisation using probes against the TRS and 4EHP genes would determine whether the spatiotemporal expression supports the phenotypes observed. Looking at published *eif4e2* data (<http://zfin.org/action/figure/all-figure-view/ZDB-PUB-040907-1?probeZdbID=ZDB-EST-041111-328>) it appears to be strongly expressed in the somatic muscle and head during ISV and Ct vessel formation, supporting the authors hypothesis.*

Response: Following the reviewer's suggestion, we investigated the expression patterns of *trs* and *4ehp* using whole-mount RNA *in situ* hybridization during zebrafish embryogenesis at 24 hpf, 48 hpf, 72 hpf, and 4.5 dpf. As the reviewer mentioned, *4ehp* was highly expressed in the developing trunk at 24 hpf (Supplementary Fig. 8e, e'), consistent with the ISV defects upon *4ehp* knock-down (Supplementary Fig. 7d). In addition, the expression of both *trs* and *4ehp* was also detected in the developing hindbrain during the period in which CtAs in the hindbrain were actively formed, further supporting their involvement in cerebrovascular formation. We have described the correlation of the vascular phenotypes (ISVs in the trunk and CtAs in the hindbrain) and their spatiotemporal expression patterns in the revised manuscript (Supplementary Fig. 8).

1. Fang P, *et al.* Structural basis for full-spectrum inhibition of translational functions on a tRNA synthetase. *Nat Commun* **6**, 6402 (2015).
2. Akabayov SR, Akabayov B, Richardson CC, Wagner G. Molecular crowding enhanced ATPase activity of the RNA helicase eIF4A correlates with compaction of its quaternary structure and association with eIF4G. *J Am Chem Soc* **135**, 10040-10047 (2013).
3. Torres-Larios A, *et al.* Structural basis of translational control by Escherichia coli threonyl tRNA synthetase. *Nat Struct Biol* **9**, 343-347 (2002).
4. Uniacke J, *et al.* An oxygen-regulated switch in the protein synthesis machinery. *Nature* **486**, 126-129 (2012).
5. Cho PF, *et al.* A new paradigm for translational control: inhibition via 5'-3' mRNA tethering by Bicoid and the eIF4E cognate 4EHP. *Cell* **121**, 411-423 (2005).
6. Cho PF, Gamberi C, Cho-Park YA, Cho-Park IB, Lasko P, Sonenberg N. Cap-dependent translational inhibition establishes two opposing morphogen gradients in Drosophila embryos. *Curr Biol* **16**, 2035-2041 (2006).
7. Han JM, *et al.* Hierarchical network between the components of the multi-tRNA synthetase complex: implications for complex formation. *J Biol Chem* **281**, 38663-38667 (2006).
8. Jeong SJ, *et al.* Inhibition of MUC1 biosynthesis via threonyl-tRNA synthetase suppresses pancreatic cancer cell migration. *Exp Mol Med* **50**, e424 (2018).
9. Braunstein S, *et al.* A hypoxia-controlled cap-dependent to cap-independent translation switch in breast cancer. *Mol Cell* **28**, 501-512 (2007).
10. Shanware NP, *et al.* Glutamine deprivation stimulates mTOR-JNK-dependent chemokine secretion. *Nat Commun* **5**, 4900 (2014).
11. Roybal CN, *et al.* Homocysteine increases the expression of vascular endothelial growth factor by a mechanism involving endoplasmic reticulum stress and transcription factor ATF4. *J Biol Chem* **279**, 14844-14852 (2004).

Reviewers' comments:

Reviewer #1 (Remarks to the Author):

The authors have improved their manuscript and they have answered extensively concerns from the previous review. This continues to be a very interesting manuscript and one with content appropriate for publication in Nature. However, this reviewer still has several concerns (in large part owing to the extensive data presented).

Major concerns

1. Line 103 "...434 proteins identified ..." An examination of the data in Figure S1 leaves open the possibility for non-direct interactions (i.e. either RNA sensitive or ribosome associated). There is no immediate ranking of these interactions given (i.e. often seen as percentage of the entire protein sequence found by mass spectrometry). At the same time, eIF4A is not identified in the screen see in Figure S1C.

2. Line 130 – If the full length TRS interacted poorly with 4EHP, is the complex that is immunoprecipitated with TRS strep a monomer or a dimer of TRS?

3. Line 222 – The identification of over 2,900 transcripts would seem to suggest a lack of specificity. If the yeast mRNA transcripts were tested, how many might be found? It would also be helpful to know the number of transcripts that were isolated using the ARS (no number is given).

4. Line 354 – Although there appears to be a TRS-eIF4A interaction seen in Figure 6, panel A, there does not appear to be an equivalent interaction seen in panel D where one would expect to see eIF4A as a subunit of eIF4F. Also, why is the amount of 4EHP reduced in the siTRS lane? Again, in panel E, in the absence of eIF4G which might allow for more TRS-eIF4A complexes, there is essentially no eIF4A found in the siEIF4G lane.

5. Line 397 – It is not clear why not all of the eIF3 subunits are pulled down together (i.e. as seen in the WCL lane in panel F of Figure 7). Do these represent free or excess subunits not in the entire factor? Or do they represent a subspecies of eIF3?

Minor comments

1. Line 36 – This should be "analogous to the eIF4F-mediated translation" rather than eIF4G.

2. Line 222 – it would be of value to know how many of the 2,928 transcripts contained elements that would be similar to the stem loop identified in the VEGF mRNA, especially in the 5' UTR. Secondly, it would ultimately be important to know the molar amount of interacting components with TRS to have some feeling for which were more likely to be found in cells. As noted in the Discussion (line 459), the timing for the expression of both 4EHP and VEGF seems to be coordinated (i.e. for day 10.5).

Reviewer #2 (Remarks to the Author):

The revised manuscript by Sunghoon Kim and coworkers is very much improved compared to the original version. While the original manuscript itself had substantial data, the authors have now performed additional experiments in response to the reviewers comments. I must say that I am overall impressed with the quantity and quality of data presented and they have addressed

satisfactorily all my comments. I therefore strongly recommend publication of the manuscript.

Rajan Sankaranarayanan

Reviewer #3 (Remarks to the Author):

Overall, I am satisfied that the authors have strong data to reveal a novel interaction between 4EHP and TRS. However, I do not feel that they have shown what the functional significance of this interaction is. Without this information, we are left with the discovery of an interesting interaction, the function of which will need to be determined in the future. Overall, I am not convinced from the revised manuscript that this is really a translation initiation complex.

Some specific comments:

No details are provided for how the ATPase data in the author rebuttal letter was generated. For example, is this RNA-dependent ATPase activity? The lack of rigor in how this data has been generated leaves me unable to assess its quality. From this data, I am not satisfied that the authors have rigorously established that TRS regulates the ATPase activity of eIF4A.

As originally stated in the first review – the interaction data rely on over-expression of proteins to test interactions. The authors state that this is a limitation of any experiments that rely on overexpression. I agree with this statement – which is why one needs to be more rigorous before generating models from overexpression studies! Other approaches should have been used to rigorously test these models – for example, inducible promoters could have been used to limit overexpression levels of proteins (perhaps equal of endogenous). The fact that the study hasn't even tried to overcome this type of limitation is a problem that lowers overall enthusiasm for this study.

The eIF4A interaction data presented in Fig S10a needs to include a control lane where TRS is not included. Without this control it is not possible to interpret the data since eIF4A may bind the resin in the conditions used. I also don't understand what "The beads were washed intensively" means in the figure legend. Please provide details about how the experiment was done – how much resin, what buffer, what volume, how many washes etc. Without this information, it is impossible to assess the rigor of the experiment and it is not possible for anyone else to replicate the data in the future.

Reviewer #4 (Remarks to the Author):

The authors have addressed all my concerns.

Responses to Reviewers` Comments

Reviewer #1 (Remarks to the Author):

The authors have improved their manuscript and they have answered extensively concerns from the previous review. This continues to be a very interesting manuscript and one with content appropriate for publication in Nature. However, this reviewer still has several concerns (in large part owing to the extensive data presented).

Response: We deeply appreciate the Reviewer #1 that he/she continuously supports our study.

Major concerns

Q1. Line 103 "...434 proteins identified ..." An examination of the data in Figure S1 leaves open the possibility for non-direct interactions (i.e. either RNA sensitive or ribosome associated). There is no immediate ranking of these interactions given (i.e. often seen as percentage of the entire protein sequence found by mass spectrometry). At the same time, eIF4A is not identified in the screen see in Figure S1C.

Response: As Reviewer #1 mentioned, we also assume that many proteins identified from the TRS-interactome analysis would be indirectly TRS-associating proteins. As the Reviewer guessed, they might possibly be either RNA sensitive or ribosome associated proteins because human TRS functions as a translation initiation factor that interacts with other translation initiation proteins, which might recruit those proteins. As requested, we have ranked gene list of the TRS-interactome as a percentage of the protein sequence found by mass spectrometry in the revised manuscript (Fig. S1b).

As pointed out by the Reviewer, we did not identify eIF4A as a TRS-interacting protein in the screen using yeast two-hybrid system (Fig. S1c). Although the two-hybrid screening system has identified many previously unknown protein-protein interactions for last decades, it cannot uncover all the potential interactors as most of other screening approaches. There are many reasons for missing the potential interactors. Mostly frequently, the cDNA library prepared for the two-hybrid screening may not cover 100% of the human genome proteins.

Besides, some proteins may not be able to fold correctly within the yeast cells and steric hindrance resulting from the fusion to the yeast proteins may inhibit the interactions between the two interactors. For this limitation, most of the primary screening results are complemented by another approaches (in our case, we did the yeast two-hybrid and affinity purification screening combined with mass spectrometry) and also further validated by manual approaches based on the hypothesis and previous knowledge.

Q2. Line 130 – If the full length TRS interacted poorly with 4EHP, is the complex that is immunoprecipitated with TRS strep a monomer or a dimer of TRS?

Response: The human TRS belongs to the class-II aminoacyl-tRNA synthetase, which forms a dimer via the catalytic domain (Fang P *et al.*, Structural basis for full-spectrum inhibition of translational functions on a tRNA synthetase. *Nat. Commun.*, 6, 6402, 2015). As we showed in the previous “Response-to-Reviewers”, we analyzed this issue using *in vitro* reconstitution of a complex of full-length TRS, 4EHP, and eIF4A (please see the figure below).

Due to the intrinsically unstable biophysical properties of TRS, mainly caused by its flexible N-terminal region including UNE-T, we conducted cross-linking to enhance the stability of the complex. As shown in the figure below, a cross-linked band corresponding to the covalent TRS-4EHP-eIF4A complex was clearly visible in the monomeric state (stoichiometry 1:1:1) and dimeric state (stoichiometry 2:2:2) following “denaturing SDS-PAGE”.

Together, these published and our results suggest that the complex of 4EHP and TRS-Strep is formed in dimeric state (stoichiometry 2:2).

Q3. Line 222 – The identification of over 2,900 transcripts would seem to suggest a lack of specificity. If the yeast mRNA transcripts were tested, how many might be found? It would also be helpful to know the number of transcripts that were isolated using the ARS (no number is given).

Response: We fully understand the Reviewer’s concern. It is known that this “omics” type of experiments (such as RIP-seq in this case) usually bear many false (positive and negative) hits and that is why the selected results are subjected to further experimental validation (which we did using cell and *in vivo* models). To minimize the potential non-specificity in our experiments, the TRS-enriched transcripts were independently subtracted by three negative control sets using AlaRS, IRS and PRS (also following the Reviewer’s previous advice). The results consistently showed the transcripts for system development in the range of 30~40% in the TRS-enriched transcripts while they normally exist in 5.8% in the whole transcripts (see the figure below). Thus, these results suggest that the transcripts for system development should be preferentially targeted by TRS.

Figure. Distribution of the human TRS-targeted (a) and whole mRNAs (b) classified by the GO terms in the Biological Process category using the Database for Annotation, Visualization and Integrated Discovery (DAVID).

Regarding the yeast mRNA transcripts, yeast might generate fewer numbers of transcripts in TRS-enriched RIP-seq analysis simply due to the lower complexity of the transcriptome. However, yeast TRS does not have the function as human TRS in the control of translation initiation. It is worth reminding that we used *Schizosaccharomyces pombe* for the interaction experiments between TRS and 4EHP, because this is the only species containing 4EHP homolog gene among yeasts.

Nevertheless, we actually tried to analyze yeast mRNA transcripts, but, in the process of preparing the experiment, we realized that it is not technically feasible at this moment because no antibody is available specific to yeast TRS that should be used to enrich TRS-bound transcripts. We wish the Reviewer's kind understanding of the results mentioned above and current technical limitations.

Lastly, we apologize for somewhat confusing schematic representation of the workflow to identify TRS-targeted mRNAs (shown in Fig. 4a). We have thus provided the workflow with the information on the number of transcripts that were isolated using the ARS in Fig. 4a in the revised manuscript. Accordingly, the workflows shown in Supplementary Fig. 3a and d have been also changed in the revised manuscript.

Q4. Line 354 – Although there appears to be a TRS-eIF4A interaction seen in Figure 6, panel A, there does not appear to be an equivalent interaction seen in panel D where one would expect to see eIF4A as a subunit of eIF4F. Also, why is the amount of 4EHP reduced in the siTRS lane? Again, in panel E, in the absence of eIF4G which might allow for more TRS-eIF4A complexes, there is essentially no eIF4A found in the siEIF4G lane.

Response: As the Reviewer pointed out, the interaction shown in Fig. 6d was not equivalent. As described in the manuscript, suppression of one complex did not appear to completely prevent eIF4A binding to the other intact complex because eIF4A is commonly associated with both TRS-4EHP and eIF4E-4G complexes. We re-performed the experiments and obtained similar results with a significant improvement.

With regard to the question of why the amount of 4EHP is reduced in the siTRS lane, we assume that the TRS may enhance the interaction of 4EHP with the cap. It has been reported that eIF4G has stimulatory effect on the cap recognition of eIF4E and dramatically enhances the interaction of eIF4E with the mRNA 5'-cap structure (Ashkan Haghghat and Nahum Sonenberg. eIF4G dramatically enhances the binding of eIF4E to the mRNA 5'-cap structure. *J. Biol. Chem.*, 272(35):21677-80, 1997). Our results also showed significantly reduced cap-binding of eIF4E when eIF4G was knocked-down. We also re-performed the experiments of Fig. 6e, and the results showed the increased TRS-eIF4A complex formation when eIF4G was silenced. Thus, the previous Fig. 6d and Fig. 6e have been replaced by these newly obtained results in the revised manuscript.

Q5. Line 397 – It is not clear why not all of the eIF3 subunits are pulled down together (i.e. as seen in the WCL lane in panel F of Figure 7). Do these represent free or excess subunits not in the entire factor? Or do they represent a subspecies of eIF3?

Response: As we previously explained in “Response-to-Reviewers”, we indeed observed that all eIF3 subunits except for eIF3A and I were immunoprecipitated with TRS in pull-down experiments following co-expression of TRS-Strep with each of the FLAG-tagged eIF3 subunits (see the results below), suggesting that TRS would interact with the entire eIF3 complex.

Like many other cases, we believe that some of the eIF3 subunits could be pulled down with direct TRS-interacting subunits. To distinguish potential direct and indirect TRS binders, we conducted *in vitro* pull-down assays with GST-TRS and each of the eIF3 subunits expressed as FLAG-tagged proteins using glutathione-Sepharose. To exclude indirect binders, we thoroughly washed the TRS-binding mixture bound to the beads, and found that the subunits eIF3B, D, F, and L survived the stringent washing step and co-precipitated with GST-TRS (Fig. 7e).

Endogenous TRS was immunoprecipitated from WI-26 cells with a mouse anti-TRS antibody and examined for associated eIF3 subunits using antibodies recognizing each of the subunits (Fig. 7f). We also performed intensive washing steps for these experiments, and again found that subunits B, D, F, L (and E) co-purified with TRS, further confirming the above results. For reference, direct eIF4G-interaction eIF3 subunits were assigned to eIF3C, D, and E (Villa *et al.*, Human eukaryotic initiation factor 4G (eIF4G) protein binds to eIF3c, -d, and -e to promote mRNA recruitment to the ribosome. *J. Biol. Chem.*, 288(46): 32932–32940, 2013).

Figure. Co-immunoprecipitation assay of TRS-Strep co-expressed with each of the FLAG-tagged eIF3 subunits in 293T cells. TRS-Strep was purified using Strep-Tactin resin, and co-precipitated eIF3 subunits were determined by immunoblotting with anti-FLAG antibody.

Minor comments

Q1. Line 36 – This should be “analogous to the eIF4F-mediated translation” rather than eIF4G.

Response: We have changed “analogous to the eIF4G-mediated translation” to “analogous to the eIF4F-mediated translation” in the revised manuscript.

Q2. Line 222 – it would be of value to know how many of the 2,928 transcripts contained

elements that would be similar to the stem loop identified in the VEGF mRNA, especially in the 5' UTR. Secondly, it would ultimately be important to know the molar amount of interacting components with TRS to have some feeling for which were more likely to be found in cells. As noted in the Discussion (line 459), the timing for the expression of both 4EHP and VEGF seems to be coordinated (i.e. for day 10.5).

Response: Since TRS appears to mainly target the transcripts for system development, we focused to the transcripts (see the Table below) involved in vascularization as the representative sample case and manually analyzed the presence of the potential TRS-binding stem and loop in the 5'UTR region using RNA structure prediction (Gruber *et al.*, Strategies for measuring evolutionary conservation of RNA secondary structures. BMC Bioinformatics 9, 122, 2008). Interestingly, all of the examined transcripts showed the presence of a potential stem and loop structure containing threonine anticodon-like bases (highlighted in red). With this high probability of the potential TRS-binding stem and loop in this set of transcripts, we can assume that the similar structure may exist in other transcripts.

As aforementioned, we think that all the components including TRS are constitutively expressed, and the assembly of the functional 4EHP-TRS complex is determined by the expression of 4EHP in a specific context. Although it is to be determined whether this complex would compete with the general translation initiation complex for the shared components such as eIF4A, PABP etc., they may exist in enough amount to independently accommodate the two complexes. As nicely commented by the Reviewer, this issue should be systematically addressed in the consideration of several parameters including the temporal factor mentioned in the Discussion in future studies.

Table. Sequences containing potential anticodon-like loop structures in the 5' UTR of mRNAs involved in vascularization

Gene symbol	Potential anticodon stem-loop sequence	Stem length (bp)
ADRB2	AGTGTGCAGGACGAGT [#] CCCCACCACACC	9
Akt1	TGCATCCTGGTCCTGTCTTCCTCATGTTT	8
ANG	CCGTGGAGGCAGTGCTCTCGCGG	8
ANG	GTCCTGCCCGTTTCTGCGGAC	6
ANG	TGGCAGATGGTGCTGTTCGACCAAGTGTCAA	10
ANXA2	GGCGCACGGCCCAAGTAAAGCGGGGCGCGCC	10

BMP4	GCAGCGCCACAGTCCCCGGCCCTCGC	8
BMP4	GAAGCTAGGTGAGTGTGGCATCCGAGCTG	9
CDX2	AACCATTGGTGTCTGTGTCATTAATA	7
CITED1	CCGAGGCCAGTCTGCTGCCGTGTGCGTCAGG	12
Col1a1	GTTTCTCCTCGGGGTCCGAGCAGGAGGC	10
Col5a1	GCCCGGGCCGTGACCCGCGC	7
EGFL7	GGGAGGCACAGGTGGCCCC	6
EGFL7	TGGCTGGGCCCGTGTGAGGGCTTCGCG	9
EGFL7	CCTCCAGGCGGCCAGTGGCCTGAGGCC	8
FGF18	CGGAGCGGCCGTGACGCTTTCG	6
HAND2	AGGAGCGAGGACAGTTACTCGCAGCT	9
HAND2	GAGCCCGAGCCGCGGTCTTCGAGCTCCAAGGCTC	12
HOXA3	ATTGGCGGCGGAGTGTACGTGACCGC	8
HOXA3	GCGGCGGAGTGTACGTGACCGC	6
JUN	GTCGTCGGAGTCCGGGCGGC	6
JUN	GCAAGAGAAGAAGGACGTGCGCTCAGCTTCGCTCGC	14
JUN	CAGGTCGGCAGTATAGTCCGAAGT	8
KDR	GCGCCGCAGAAAGTCCGTCTGGCAGC	8
MMP14	TAGGAATTCAGTTCAGTGCCTA	8
NPR1	GGGGTGAGCGTCCCCTCCGCTCCTGCTCC	10
NRTN	GGCAGGCGTTCAAAGTCAAAGGCC	6
NRTN	GGAGGGACAGACGGGGCGTGGGCTGACCATCC	13
NRTN	TGCCCCAGCGCCCTGTGCCGTTGGCTGC	8
OVOL2	GGCGAGGCGGGAAAGTGGGCTGTGGCCGCC	10
OVOL2	TCGCCAGGCGTGGGGG	6
PAX2	AGTGGCAAGTGGCGGCTACT	6
PAX2	CTCAAGTCTGAAGTTGAGTTT	8
PDGFA	GCGCCGCGGAGGGGTGCTGGGCCGCGCT	11
PDGFA	CGAGGCCCGGGCGGGGTGGTGGCTGCCAGGCGGCTCG	14
PGF	GCTCGGGAACCTCTGCGGTGGGC	7
PLAU	CCCAGCCTGCGGGCATCTGGTAGATGAAGCTTGCTTGGG	14
PTPRJ	AGCCGCGCGCTGGGGGTGGGCGCCGCTCGCT	11
RIPK2	GGCGCCTGAGCGCGGCGTGGGAGCCTGGGAGCC	12
SIGIRR	GAAGCCTCTGACCTGTCCAGGTGCCCTGT	7
SMO	GCCTCCGCGGCCGCCGAGGTGCGTGTGGCCGGGGGGC	15
SMO	GCCTCCGCGGCCGCCGAGGTGCGTGTGGCCGGGGGGG	14
TICAM1	GCGCGCTACGGTCCGCGGGC	6
VEGFA	GGTGAGGCGGGCGGTGTGCGCAGACAGTG	7

#, Threonine anticodon-like bases are highlighted in red.

Reviewer #3 (Remarks to the Author):

Overall, I am satisfied that the authors have strong data to reveal a novel interaction between 4EHP and TRS. However, I do not feel that they have shown what the functional significance of this interaction is. Without this information, we are left with the discovery of an interesting interaction, the function of which will need to be determined in the future. Overall, I am not convinced from the revised manuscript that this is really a translation initiation complex.

Response: The entire data shown in our work are converged to show the functional significance for translation initiation of the interaction between TRS and 4EHP. In addition to comparative structural and biochemical analyses, we proved the function of this complex using molecular and cell-based assays as well as *in vivo* animal models at the level that current techniques allow.

To our knowledge, a few translation initiation complexes containing 4EHP have been reported. Some representative complexes include the 4EHP-RNA-binding protein RBM4-HIF-2 α complex in the oxygen tension-specific translation initiation (Uniacke J et al., An oxygen-regulated switch in the protein synthesis machinery. *Nature* 486(7401):126-9, 2012), the 4EHP-Bicoid protein complex in the translational repression of caudal mRNA (Cho PF, et al., A new paradigm for translational control: inhibition via 5'-3' mRNA tethering by Bicoid and the eIF4E cognate 4EHP. *Cell* 121(3):411-23, 2005), and the 4EHP-Grb10-interacting GYF protein 2-zinc finger protein 598 complex in the translation repression of mRNAs during embryonic development (Morita M et al., *Mol Cell Biol*, A novel 4EHP-GIGYF2 translational repressor complex is essential for mammalian development. 32(17):3585-93, 2012; Peter D et al., GIGYF1/2 proteins use auxiliary sequences to selectively bind to 4EHP and repress target mRNA expression. *Genes Dev*, 31(11):1147-1161, 2017). Based on the structural similarity of the 4EHP-TRS complex with the other translation initiation complexes and the related activities in cell and *in vivo*, it is difficult to consider that the 4EHP-TRS complex would not be involved in translation control.

Some specific comments:

Q1. No details are provided for how the ATPase data in the author rebuttal letter was generated. For example, is this RNA-dependent ATPase activity? The lack of rigor in how

this data has been generated leaves me unable to assess its quality. From this data, I am not satisfied that the authors have rigorously established that TRS regulates the ATPase activity of eIF4A.

Response: We acknowledge that the ATPase data would need more detailed description. We found that we also made typos in the figure for ATPase data. The amounts of proteins we used for the assays were not 20 μM or 40 μM for each reaction, but 2 μM or 4 μM (see the figure below). We sincerely apologize for our carelessness and are much thankful the Reviewer for pointing this.

ATP hydrolysis by purified eIF4A was measured using a Malachite Green Phosphate Kit (R&D Systems, Inc., USA). For all experiments, the specific activity was determined using the standard curve generated from a phosphate standard (R&D Systems, Inc., USA). The reaction mixture (20 μl) contains 2 μM or 4 μM eIF4A, 100 μM ATP, and indicated concentration of TRS or LRS (leucyl-tRNA synthetase as a negative control) in a buffer containing 50 mM Tris-HCl, pH 8.0, 50 mM KCl, 2 mM MgCl_2 , 0.1 mg/ml bovine serum albumin, and 1 mM DTT. The reaction was incubated at 37 $^\circ\text{C}$ for 30 min and the aliquots (5 μl) were taken for inorganic phosphate (Pi) determination. The reaction was stopped by the addition of 10 μl of malachite green reagent A (ammonium molybdate in 3 M sulfuric acid) followed by 10 μl of malachite green reagent B (malachite green oxalate and polyvinyl alcohol) to the mixture, and the mixture was incubated at room temperature for 10 min before an optical density was taken using a microplate reader set to 620 nm, according to the manufacturer's instruction. All experiments were performed in triplicate.

Figure. Enhancement of the ATPase activity of eIF4A by TRS. The effect of TRS on the ATPase activity of eIF4A was examined at 37 °C for 30 min. The amounts of released Pi were determined using a Malachite Green Phosphate Kit (R&D Systems, Inc., USA). Values are means \pm SEM of three independent experiments.

Q2. As originally stated in the first review – the interaction data rely on over-expression of proteins to test interactions. The authors state that this is a limitation of any experiments that rely on overexpression. I agree with this statement – which is why one needs to be more rigorous before generating models from overexpression studies! Other approaches should have been used to rigorously test these models – for example, inducible promoters could have been used to limit overexpression levels of proteins (perhaps equal of endogenous). The fact that the study hasn't even tried to overcome this type of limitation is a problem that lowers overall enthusiasm for this study.

Response: We agree the Reviewer's point. We accordingly used a doxycycline (Dox)-regulated TRS inducible cell line (Jeong et al., Inhibition of MUC1 biosynthesis via threonyl-tRNA synthetase suppresses pancreatic cancer cell migration. *Exp. Mol. Med.*, 50, e424, 2018) to confirm that the TRS interaction with 4EHP regulates VEGF expression.

Briefly, MIA PaCa-2 cells were cultured in DMEM supplemented with 10% FBS and antibiotics and seeded evenly in a 60 mm dish and incubated for 12 h to reach ~90% confluence. When the cells were ready for transfection, 2 μ l of Myc-tagged TRS lentiviral particles and 3 μ l of 10 mg/ml polybrene were supplemented with 2 ml of complete media and added to the plate. After 16 h of incubation, the culture media was replaced with 3 ml of fresh complete media containing 1 μ g/ml puromycin and incubated for an additional 48 h. The cells were gradually selected by treating with puromycin every 48 h. The efficiency of TRS overexpression was checked using the cells treated with 2.5 μ g/ml of doxycycline (Dox) for the indicated times, followed by immunodetection.

MIA PaCa-2 cells that expressed inducible Myc-tagged TRS were cultured in the presence or absence of Dox for 24 to 48 h. Myc-tagged TRS was immunoprecipitated with anti-Myc antibody, and co-precipitation of 4EHP was determined by immunoblotting with 4EHP antibody. Levels of VEGF in the cell culture supernatants were determined by ELISA.

As seen in the figure below, Myc-tagged TRS showed Dox-dependent expression. Immunoprecipitation using anti-Myc antibody followed by Western blot revealed that the

interaction of 4EHP with TRS was increased in a Dox-dependent manner. VEGF levels were also increased in a Dox-dependent manner. Thus, these results further indicate that the TRS interaction with 4EHP positively regulates protein synthesis.

Figure. The interaction of 4EHP with TRS in a Dox-dependent manner (**a**). MIA PaCa-2 cells that expressed inducible Myc-tagged TRS were cultured in the presence or absence (-) of Dox for 24 to 48 h. Myc-tagged TRS was immunoprecipitated with anti-Myc antibody, and co-precipitation of 4EHP was determined by immunoblotting with 4EHP antibody. VEGF protein levels in the culture supernatants were determined by ELISA (**b**). Values are means \pm SEM of three independent experiments.

Q3. The eIF4A interaction data presented in Fig S10a needs to include a control lane where TRS is not included. Without this control it is not possible to interpret the data since eIF4A may bind the resin in the conditions used. I also don't understand what "The beads were washed intensively" means in the figure legend. Please provide details about how the experiment was done – how much resin, what buffer, what volume, how many washes etc. Without this information, it is impossible to assess the rigor of the experiment and it is not possible for anyone else to replicate the data in the future.

Response: As requested, we re-performed the interaction experiment and the result has been changed in the revised manuscript. Regarding the experiment method for the interaction, we described the details in "In vitro binding assay" of the "Methods" section to be comparable to other figure legends in the previous manuscript.

REVIEWERS' COMMENTS:

Reviewer #1 (Remarks to the Author):

The authors have addressed the primary concerns of this reviewer and as such, this manuscript should likely be accepted for publication. That said, several comments are offered below for consideration by the authors. The only possible correction needed is for page 17 (Figure 7h). In keeping with the action of eIF4F, it would seem more reasonable to form the TRS•4EHP•eIF4A complex prior to binding to the m7G cap of the mRNA rather than the sequence in lines 412, 413 which suggest the association of eIF4A following the binding of the TRS•4EHP complex to the mRNA.

Minor comments

1. In this reviewer's print out, "x" showed up in a number of places which for the pdf were blanks. Examples would be line 124 (UNE-T, residues 1x80) or line 127 Kd of 2.48 xM. I believe the x was to be a dash (-) in the first case and micromolar (μ M) in the second. This may reflect the use of a Mac on the part of the reviewer. However, this error showed up frequently (see also line 634 where 4xC is likely supposed to be 4o C).

2. Line 333 – "... indicating that TRS associates with cap-bound 4EHP." The authors should note that it would also be possible for eIF4A to be present in the complex, but it was not assayed for (see supplementary Figure 10a). It might also be noted that the level of 4EHP in whole cell lysates for WI-26 and VSMC is much less than in the other panels.

3. Supplementary Figure 1d – it is still not clear to this reviewer why so little 4EHP is pulled down in the TRS-Strep/4EHP lane considering how much was present as seen in input.

In sum, this manuscript adds a new pathway to protein synthesis initiation. Although this may easily be a minor pathway such as seen with IRES-mediated initiation or re-initiation, it none the less appears to be important. It is anticipated by this reviewer that additional pathways will emerge as reflects either development or the ability of cells to resist stress and maintain homeostasis.

Reviewer #3 (Remarks to the Author):

I am satisfied that the authors have addressed my original comments and feel that the manuscript is improved and ready for publication.

Responses to Reviewers` Comments

Reviewer #1 (Remarks to the Author):

The authors have addressed the primary concerns of this reviewer and as such, this manuscript should likely be accepted for publication. That said, several comments are offered below for consideration by the authors. The only possible correction needed is for page 17 (Figure 7h). In keeping with the action of eIF4F, it would seem more reasonable to form the TRS•4EHP•eIF4A complex prior to binding to the m7G cap of the mRNA rather than the sequence in lines 412, 413 which suggest the association of eIF4A following the binding of the TRS•4EHP complex to the mRNA.

Response: We thank the reviewer for his/her insightful comments. As suggested, we have changed the sequence of the assembly of the TRS-mediated translation initiation machinery in the revised manuscript.

Minor comments

Q1. In this reviewer's print out, "x" showed up in a number of places which for the pdf were blanks. Examples would be line 124 (UNE-T, residues 1x80) or line 127 Kd of 2.48 xM. I believe the x was to be a dash (-) in the first case and micromolar (M) in the second. This error may reflect the use of a Mac on the part of the reviewer. However, this error showed up frequently (see also line 634 where 4xC is likely supposed to be 4o C).

Response: We think that these errors were caused by compatibility problems.

Q2. Line 333 – "... indicating that TRS associates with cap-bound 4EHP." The authors should note that it would also be possible for eIF4A to be present in the complex, but it was not assayed for (see supplementary Figure 10a). It might also be noted that the level of 4EHP in whole cell lysates for WI-26 and VSMC is much less than in the other panels.

Response: This issue (Supplementary Fig. 9) was indeed commented by the Reviewer #1 in the first review and addressed in the “Description of Responses-to-Reviewers” in the first revision as following:

Q8. In supplementary figure 7 it is shown that m7GTP-Sepharose can pulldown TRS (via its interaction with 4EHP). What other proteins are also pulled down with 4EHP under these conditions?

Response: When we performed experiments related to the results shown in Supplementary figure 7, we had no idea whether TRS functions similarly to eIF4G of eIF4F, and therefore did not know exactly what to check for in the pull-down samples. eIF4A was later examined as a component of the TRS-mediated complex, and in subsequent m7GTP-Sepharose pull-down experiments, we showed that TRS and 4EHP form a distinct complex separate from the eIF4G-eIF4E complex, and both complexes contain eIF4A (Fig. 6d, e). Although we acknowledge the Reviewer’s comments, given the order of our experiments and discoveries in our study, we believe it is better to keep the current dataset. However, to strengthen the original results, we have also checked AlaRS and KRS (as negative controls) and eIF4E (as a positive control) in the cap-dependent complex consisting of 4EHP and TRS, and added the new data to the revised manuscript (Supplementary Fig. 9).

Regarding the level of 4EHP in whole cell lysates for WI-26 and VSMC showing much less than in the other cells in Supplementary Fig. 9, we think that it might be depending on cell type.

Q3. Supplementary Figure 1d – it is still not clear to this reviewer why so little 4EHP is pulled down in the TRS-Strep/4EHP lane considering how much was present as seen in input.

Response: As we described in the manuscript, unlike the isolated UNE-T, the N-terminal UNE-T in the dimeric form of full-length TRS may not be fully exposed for the interaction with 4EHP *in vitro*, resulting in the weak interaction with 4EHP (Supplementary Fig. 1d). Perhaps, formation of the cellular TRS-4EHP complex might involve a conformation change of the full-length TRS to fully expose UNE-T.

In sum, this manuscript adds a new pathway to protein synthesis initiation. Although this may easily be a minor pathway such as seen with IRES-mediated initiation or re-initiation, it none the less appears to be important. It is anticipated by this reviewer that additional pathways will emerge as reflects either development or the ability of cells to resist stress and maintain homeostasis.

Reviewer #3 (Remarks to the Author):

I am satisfied that the authors have addressed my original comments and feel that the manuscript is improved and ready for publication.